# Path-Coupled Bellman Flows for
# Distributional Reinforcement Learning

**Boyang Xu** [1]  **Qing Zou** [1]  **Siqin Yang** [1]  **Hao Yan** [1]

## Abstract

Distributional reinforcement learning (DRL) models the full return distribution, but typically relies on finite-dimensional categorical or quantile approximations, often involving projection or quantile-regression approximations to the Bellman target, together with independently sampled bootstrap targets that obscure transport structure and add variance. We present Path-Coupled Bellman Flows (PCBF), a continuous-time DRL method that encodes Bellman endpoint consistency and pathwise Bellman-coupled geometry within generative flow trajectories. PCBF represents return distributions via flow matching and couples the paths of consecutive states through shared base noise, yielding a geometric Bellman flow relation between velocity fields. This structure enables a $\lambda$-parameterized control-variate target that reduces training variance while preserving the source and Bellman endpoint geometry. Experiments on analytically tractable MRPs, OG-Bench, and D4RL show improved distributional fidelity, training stability, and competitive offline RL performance.

**The code is available at:** https://github.com/BoyangASU/path-coupled-bellman-flows

## 1. Introduction

Distributional reinforcement learning (DRL) (Bellemare et al., 2017) models the full distribution of returns rather than only their expectation, enabling richer representations of uncertainty and often leading to improved empirical performance. Despite this success, most practical DRL algorithms typically rely on finite-dimensional approximations

[1] School of Computing and Augmented Intelligence, Arizona State University, Tempe, AZ, USA. Correspondence to: Hao Yan <haoyan@asu.edu>.

*Proceedings of the 43rd International Conference on Machine Learning*, Seoul, South Korea. PMLR 306, 2026. Copyright 2026 by the author(s).

over fixed supports (Bellemare et al., 2017) or quantile assignments (Dabney et al., 2018b). However, these discretization strategies introduce inherent limitations. Because the Bellman update rarely aligns with fixed support points or quantile locations, these methods rely on heuristic projection steps that introduce bias and limit the expressivity of the learned distribution.

To overcome the limitations of discrete projections, it is natural to reframe DRL as a problem of continuous probability transport. Fundamentally, the distributional Bellman equation defines an affine transport relationship: the return distribution at the current state is a direct transformation of the successor state's distribution. This perspective makes flow matching (Lipman et al., 2023) a highly attractive framework. By learning a continuous neural velocity field that transports samples from a simple base prior (e.g., Gaussian noise) to a complex target distribution, flow matching sidesteps heuristic projections and gracefully models the geometry of the Bellman update.

Yet, directly enforcing an uncorrected pointwise Bellman map inside flow composition fails in two critical ways. First, it violates the initial boundary condition: flow matching requires the generation process to start from a fixed simple prior, such as a standard Gaussian. However, an uncorrected Bellman update shifts this starting distribution through the reward and discounted successor value, making the standard flow-matching objective ill-posed. Second, when the noise driving the current and successor distributions is sampled independently, their intermediate trajectories are not pathwise aligned. As a result, Bellman consistency can only be enforced at the endpoint, leading to high-variance per-sample targets that destabilize critic learning.

In this work, we propose **Path-Coupled Bellman Flows (PCBF)**, which encodes Bellman endpoint consistency and pathwise Bellman-coupled geometry within generative flow trajectories while addressing both issues through a source-consistent Bellman path correction and shared-noise path coupling. Instead of applying the Bellman update directly to intermediate flow states, PCBF repairs the flow path so that it starts from the required base prior while still ending at the Bellman target. This separates the geometric requirement of flow matching from the stochasticity of Bellman boot-

strapping. On top of this corrected geometry, PCBF couples current and successor return flows using shared base noise, aligning their intermediate trajectories rather than enforcing Bellman consistency only at the endpoint. This pathwise structure induces a $\lambda$-parameterized family of training targets that interpolates between direct sample-based Bellman supervision and variance-reduced supervision using successor-flow velocity predictions. Thus, PCBF cleanly separates Bellman path correction from variance control.

Theoretically, we characterize the population-optimal velocity field induced by the PCBF path, analyze the bias–variance behavior of the $\lambda$-target, and show that shared-noise Bellman generator updates inherit the standard $\gamma$-contraction, with an additional $t$-factor contraction along PCBF interpolants. Empirically, PCBF accurately recovers ground-truth return distributions on toy MRPs and achieves competitive performance on OGBench (Park et al., 2025a) and D4RL Adroit (Fu et al., 2020).

**Contributions.** Our main contributions are: **(i)** a source-consistent Bellman-interpolated path that resolves the $t{=}0$ boundary mismatch of uncorrected pointwise Bellman paths while preserving the Bellman endpoint at $t{=}1$; **(ii)** a shared-noise path coupling that aligns current and successor return flows; **(iii)** a $\lambda$-parameterized control-variate target, with a distribution-free $L_2$ bias bound and a linear–Gaussian closed form showing why shared-noise coupling reduces intrinsic bias; **(iv)** a comprehensive empirical evaluation on analytically tractable return laws, OGBench, and D4RL.

**Conflict of Interest Disclosure** The authors declare that they have no financial or non-financial competing interests that could have influenced the work reported in this paper.

## 2. Related Work

### 2.1. Generative Models in Offline RL

Generative models have become prominent in offline RL for trajectory planning, behavior modeling, and policy extraction, including diffusion-based planners and policies (Janner et al., 2022; Wang et al., 2023; Hansen-Estruch et al., 2023) and flow- or energy-guided policy methods (Chen et al., 2024; Lu et al., 2023; Zhang et al., 2025). These works primarily improve the actor or planner. PCBF instead focuses on the critic side: learning expressive and stable return distributions that can guide offline policy extraction.

### 2.2. Distributional Reinforcement Learning

To capture the full complexity of cumulative returns, the DRL paradigm shifts the focus from estimating expected scalar values to modeling the entire return distribution (Bellemare et al., 2017). Classical approaches initially re-

lied on categorical projections (Bellemare et al., 2017) and quantile regression (Dabney et al., 2018b) for discrete action spaces. This framework was later successfully scaled to continuous control domains by D4PG (Barth-Maron et al., 2018), which integrated categorical distributional critics with deterministic policy gradients, and was further advanced by continuous variants like DSAC (Duan et al., 2021). Despite their success, these traditional methods are inherently constrained by discrete projections, fixed support boundaries, or moment-matching approximations (Nguyen-Tang et al., 2021), fundamentally limiting continuous critic expressivity. To overcome these limits and match the continuous nature of modern generative actors, recent efforts have attempted to unify DRL with continuous flow matching.

### 2.3. Flow Matching for Value Function Modeling

Beyond policy and trajectory generation, generative models are increasingly applied to other RL components, especially value and return modeling.

**Scalar value and reward flow models.** EVOR (Espinosa-Dice et al., 2025) performs inference-time policy extraction using a distributional reward model learned via standard flow matching. Approaches such as floQ (Agrawalla et al., 2026) use flow matching to parameterize value functions, mapping noise to scalar value estimates to enable iterative compute scaling.

**Temporal-difference flow matching.** TD-Flow (Farebrother et al., 2025) combines temporal-difference bootstrapping with flow matching over probability paths. When applied to rewards, however, TD-Flow naturally corresponds to a return distribution for a transformed random-horizon MDP, where the process terminates at each step with probability $1 - \gamma$. This differs from the conventional discounted return distribution usually studied in DRL: as discussed by Bellemare et al. (2023), the two interpretations agree in expectation but generally differ as distributions. In contrast, PCBF is designed to model the standard discounted return distribution while preserving its Bellman geometry within flow matching.

**Endpoint Bellman alignment.** Concurrent methods such as DFC (Chen et al., 2025) and Bellman Diffusion (Li et al., 2024) attempt to align Bellman target distributions but rely on independently sampled noise. Such endpoint-level matching lacks pathwise coupling, which can lead to misaligned vector fields and high-variance training targets.

**Self-consistency and source boundaries.** Value Flows (Dong et al., 2026) uses flow matching to model return distributions, combining endpoint supervision (BCFM) with a DCFM-style self-consistency term built

from $r + \gamma z'_t$ at each $t$. A full-$t$ DCFM-style Bellman density constraint can tension with the Gaussian source at $t = 0$, but the practical objective mitigates this via BCFM anchoring and weighting. We revisit this distinction, with an expanded comparison to PCBF, in Appendix B.

**Quantile-based coupling.** FlowIQN (Groom et al., 2026) identifies a projection-metric mismatch in flow-matching distributional critics: independently paired source and Bellman-target samples yield a valid CFM regression objective, but not necessarily a Wasserstein-aligned Bellman projection. FlowIQN addresses this issue in one-dimensional return space by sorting source quantiles and Bellman-target samples within each minibatch, thereby approximating the monotone optimal-transport coupling and producing a Wasserstein-aligned projection objective. This direction is complementary to PCBF: FlowIQN focuses on the endpoint source–target coupling used to project Bellman target laws, whereas PCBF focuses on the time-continuous Bellman path itself—repairing the source boundary at $t = 0$, coupling current and successor flows through shared noise, and using a $\lambda$-parameterized control variate to reduce successor-bootstrap variance.

Building on the unified flow-matching framework of Lipman et al. (2024), PCBF addresses these issues through source-consistent Bellman path repair and shared-noise path coupling. Together with its control-variate training target, PCBF mitigates high-variance Bellman flow-matching updates and provides a stable flow-based distributional critic for offline RL.

## 3. Preliminaries

**Distributional RL.** We consider a Markov decision process (MDP) (Sutton et al., 1998) defined by the tuple $(\mathcal{S}, \mathcal{A}, p, \mathcal{R}, \gamma)$, with state space $\mathcal{S}$ and action space $\mathcal{A}$, transition kernel $p(s' \mid s, a)$, and discount $\gamma \in (0, 1)$. At each step we observe a reward sample $R$ drawn from a reward kernel $\mathcal{R}(\cdot \mid s, a, s')$ (deterministic rewards are the special case $R = r(s, a)$ a.s.). Let $\pi(a \mid s)$ denote a stochastic policy. In distributional reinforcement learning (DRL), the return is modeled as a random variable $Z^\pi(s, a)$. The evolution of return distributions is governed by the *distributional Bellman equation*

$$Z^\pi(s, a) \stackrel{d}{=} R(s, a) + \gamma Z^\pi(S', A'), \qquad (1)$$

where $(S', A')$ are random variables sampled conditional on $(s, a)$, and $\stackrel{d}{=}$ denotes equality in distribution.

**Flow matching.** Flow matching (Lipman et al., 2023) is a class of continuous-time generative models that learns a time-dependent vector field to transport samples from a

simple base distribution to a target data distribution. In contrast to denoising diffusion models (Ho et al., 2020; Song et al., 2021), which rely on stochastic differential equations (SDEs) and iterative noise injection, flow matching models are based on ordinary differential equations (ODEs), enabling simpler training and faster inference, while often achieving comparable or superior sample quality.

Given a target distribution $p$ on $\mathbb{R}^d$, flow matching learns the parameters $\theta$ of a time-dependent velocity field $v_\theta$ such that the induced flow $\psi_\theta$, defined as the solution to the ordinary differential equation

$$\frac{d}{dt}\psi_\theta(t, x) = v_\theta(t, \psi_\theta(t, x)) \qquad (2)$$

satisfies the following boundary behavior: when initialized from a simple base distribution at $t = 0$ (e.g., a standard Gaussian), the resulting dynamics transport samples so that the distribution at $t = 1$ matches $p$.

In this work, we adopt flow matching based on linear interpolation paths and uniform time sampling (Lipman et al., 2023). Specifically, given samples $X_0 \sim \mathcal{N}(0, I_d)$ and $X_1 \sim p(x_1)$, we define the linear interpolation

$$X_t = (1 - t)X_0 + tX_1. \qquad (3)$$

The flow matching objective minimizes the squared error between the learned velocity field and the ground-truth displacement direction

$$\min_\theta \ \mathbb{E}_{\substack{X_0 \sim \mathcal{N}(0, I_d) \\ X_1 \sim p(x_1) \\ t \sim \mathrm{Unif}([0,1])}} \left[ \|v_\theta(t, X_t) - (X_1 - X_0)\|_2^2 \right]. \qquad (4)$$

Intuitively, this objective trains the velocity field $v_\theta$ to predict the average direction that transports samples from the base distribution toward the data distribution along straight-line paths. At convergence, the resulting vector field defines an ODE whose solution generates samples from $p(x)$ when integrated from $t = 0$ to $t = 1$. At inference time, new samples are generated by solving the ODE in Equation (2). In this work, we use the explicit Euler method, which we find to be sufficient in practice. We refer readers to Lipman et al. (2023) for further theoretical and practical details.

## 4. Path-Coupled Bellman Flows

We now introduce *Path-Coupled Bellman Flows*, a continuous-time distributional reinforcement learning framework that integrates flow matching with the recursive geometry of the distributional Bellman equation. Our goal is to learn return distributions with flow matching while using **source-consistent Bellman-coupled paths**: the current path starts from the required base prior at $t{=}0$, reaches the Bellman target at $t{=}1$, and maintains a pathwise affine relation to the successor flow at intermediate times—without requiring every time-$t$ marginal to be a distributional Bellman

fixed point. Figure 1 illustrates the key geometric distinction. PCBF couples the current and successor return flows through shared base noise while preserving the prescribed source distribution at $t = 0$ and the Bellman endpoint at $t = 1$. In contrast, directly imposing a Bellman relation at every intermediate time can shift the source away from the base prior, while uncoupled flow critics align only endpoint samples and do not enforce pathwise Bellman geometry.

### 4.1. Boundary Mismatch of Pointwise Bellman Path

Write the successor linear flow (shared base noise $X_0$, terminal return $X'$) as

$$Z_t' = (1 - t)X_0 + tX'. \tag{5}$$

A natural attempt is to enforce the Bellman affine map *pointwise at every flow time* by defining the *uncorrected pointwise Bellman path*

$$Z_t^D \;:=\; R + \gamma Z_t'. \tag{6}$$

This construction reaches the desired Bellman endpoint, $Z_1^D = R + \gamma X'$, but it violates the source boundary required by flow matching: $Z_0^D = R + \gamma X_0$, which is generally not equal to the prescribed base noise $X_0 \sim \mathcal{N}(0, 1)$. Thus $Z_t^D$ is not a valid flow-matching path for a model whose source distribution is fixed to the base prior.

This boundary failure also clarifies how prior Bellman flow-matching objectives connect to pointwise supervision. In particular, the DCFM-style term in Value Flows (Dong et al., 2026) can be read as imposing intermediate-time consistency in velocity space: given an intermediate point $Z_t$, it evaluates the successor velocity at the Bellman-inverse point $(Z_t - R)/\gamma$ and aligns the two velocity fields. The Value-Flows DCFM term can be viewed as a same-time self-consistency penalty evaluated at the Bellman-inverse point. Unlike the exact path derivative of $Z_t = R + \gamma Z_t'$, this objective does not include the explicit $\gamma$ velocity scaling; this distinction is one source of mismatch with the affine Bellman path geometry. A literal full-$t$ DCFM-style Bellman density constraint inherits this source-boundary tension: a path satisfying $Z_t = R + \gamma Z_t'$ at every $t$ would reach the Bellman endpoint at $t=1$ but start from $R + \gamma X_0$ rather than $X_0$. The practical Value Flows objective mitigates this tension by adding BCFM anchoring and uncertainty weighting. Appendix B revisits this issue at the distributional level and records formal statements (including a degeneracy analysis for unscaled DCFM).

**Lemma 4.1** (DCFM as Bellman-inverse same-time self--consistency)**.** *The geometry-consistent DCFM objective evaluates the successor velocity at the Bellman-inverse point $Z_t' = (Z_t - R)/\gamma$ and penalizes same-time velocity mismatch with the affine scaling: $\mathcal{L}_{\text{DCFM}}(v, v_k) =$*

$$\mathbb{E}\left[\left(v(t, Z_t \mid s, a) - \gamma\, v_k\left(t, \tfrac{Z_t - R}{\gamma} \,\Big|\, s', a'\right)\right)^2\right], \text{ where } v_k$$

*denotes the lagged target-network velocity field. This objective matches the path-derivative scaling implied by $Z_t = R + \gamma Z_t'$.*

Appendix B expands on Value Flows (DCFM/BCFM objectives), explains the source-boundary tension under a literal full-$t$ DCFM constraint and how the practical objective mitigates it, and contrasts this with PCBF's source-consistent coupled-path construction.

**Path-coupled Bellman flows.** PCBF uses *path coupling*: shared base noise ties the current and successor flows so that their velocities satisfy a Bellman-shaped geometric relation while respecting the flow source at $t=0$ and the Bellman endpoint at $t=1$.

Building on this coupling, PCBF derives a family of training targets parameterized by $\lambda$. Different choices of $\lambda$ interpolate between purely sample-based Bellman supervision and variance-reduced targets that incorporate model-predicted successor dynamics. $\lambda = 0$ is unbiased; $\lambda > 0$ trades bias for variance reduction. With shared-noise coupling, the induced bias is small.

### 4.2. Bellman-Consistent Shared-Noise Paths

We begin by formalizing the pathwise structure implied by the distributional Bellman equation (1), which relates the return distributions at $(s, a)$ and $(s', a')$ through an affine transformation. Rather than enforcing Bellman structure only at isolated endpoint samples, PCBF encodes a pathwise affine coupling between current and successor flows.

**Flow-map notation.** We write $\psi_\theta^1(x_0 \mid s, a)$ for the solution at time $t=1$ of the ODE (2) with initial condition $x_0$ at $t=0$ and conditioning $(s, a)$; equivalently this is the previous "$\psi_\theta(X_0 \mid s, a, 1)$" notation. For a base noise draw $X_0 \sim \mathcal{N}(0, I_d)$, the random variable $\psi_\theta^1(X_0 \mid s, a)$ follows the learned return law $\hat{Z}_\theta(s, a)$.

To couple the current and successor flows, we generate the successor return using the *same* base noise and a slowly updated target network $v_{\theta^-}$, with flow map $\psi_{\theta^-}^1$:

$$X' = \psi_{\theta^-}^1(X_0 \mid s', a'). \tag{7}$$

The theory permits any coupling between $X_0$ and $X_1'$ with the correct terminal marginal, while the implemented PCBF algorithm uses the deterministic target-flow coupling $X_1' = \psi_{\theta^-}^1(X_0 \mid S', A')$.

We then define two time-synchronized linear interpolation

*Figure 1.* **Path-coupled Bellman geometry.** Each panel shows a single current (blue) and successor (orange) return flow, differing only in how the per-path source noise is drawn. **(a)** Uncoupled: the two flows start from independent noise, so they are unrelated except in distribution. **(b)** Source-inconsistent: the flows are paired, but the successor starts from the shifted point $R + \gamma X_0$, violating the base prior at $t = 0$. **(c)** PCBF: a single shared noise drives both flows, preserving the base prior at $t = 0$ and the Bellman endpoint at $t = 1$.

paths as follows:

$$Z_t^{s'} = (1 - t)X_0 + tX', \qquad \text{(successor path)}, \quad (8)$$

$$Z_t^s = (1 - t)X_0 + t(R + \gamma X'), \quad \text{(current path)}. \quad (9)$$

Both paths originate from the same noise realization at $t = 0$ and terminate at Bellman-related samples at $t = 1$. An equivalent expression of Equation (9) that explicitly reveals the Bellman geometry is:

$$Z_t^s = tR + \gamma Z_t^{s'} + (1 - t)(1 - \gamma)X_0. \quad (10)$$

The final term acts as a residual anchor that guarantees the exact alignment of both paths at $t = 0$, ensuring shared-noise initialization regardless of $\gamma$. At $t = 1$, the construction satisfies the distributional Bellman boundary condition $Z_1^s = R + \gamma X'$.

Differentiating (10) yields $\dot{Z}_t^s = R - (1 - \gamma)X_0 + \gamma \dot{Z}_t^{s'}$. Substituting $v_{\theta^-}(t, Z_t^{s'} \mid s', a')$ for $\dot{Z}_t^{s'}$ gives the geometric target at $\lambda = \gamma$, revealing that $\lambda = 0$ uses pure samples while $\lambda = \gamma$ eliminates $X'$ via velocity prediction.

Differentiating the current-state path in Equation (9) yields the ideal Bellman-consistent velocity

$$\dot{Z}_t^s = R + \gamma X' - X_0 =: Y. \quad (11)$$

This quantity corresponds to the BCFM target. While unbiased, $Y$ depends directly on the sampled successor return $X' = x'$ and thus suffers from high variance.

### 4.3. $\lambda$-Parameterized Control Variates

PCBF reduces this variance by exploiting information available along the successor path. Evaluating the target velocity

field along $z_t^{s'}$ yields

$$c_t = v_{\theta^-}(t, Z_t^{s'} \mid s', a'). \quad (12)$$

For linear interpolation, the true successor-path velocity is constant and equal to $X' - X_0$. We therefore define a control variate $C_t = c_t - (X' - X_0)$, which captures the discrepancy between the model-predicted successor velocity and the sample-based velocity. For the population-optimal successor field $\bar{v}^\star$, the intrinsic piece of $C_t$ has mean zero conditional on the successor interpolant $Z_t^{s'}$ (see Section 5); for learned $\bar{v}$ the residual remains random but acts as a control variate.

Using this control variate, we define the PCBF training target as follows:

$$u_t^\lambda := (R + \gamma X' - X_0) + \lambda \left[ v_{\theta^-}(t, Z_t^{s'} \mid s', a') - (X' - X_0) \right]. \quad (13)$$

Setting $\lambda = 0$ recovers the baseline BCFM estimator (unbiased, high variance). Nonzero $\lambda$ introduces a variance-reducing correction at the cost of potential bias. Early in training, $\lambda \approx \gamma$ is often effective, as it replaces the noisy successor sample with smoother velocity predictions. As the target network improves, the correction becomes a more reliable control variate: its conditional mean under successor-path information approaches that of $\bar{v}^\star$ while still providing variance reduction.

### 4.4. Policy Extraction for Offline Control

At deployment, we extract actions using a candidate-action protocol based on mean terminal returns under the learned flow; details are given in Appendix D.

**Training procedure.** Putting the above components together, PCBF trains a current velocity network $v_\theta$ on a fixed offline dataset $\mathcal{D}$, together with a slowly updated target velocity network $v_{\theta^-}$ maintained by Polyak averaging. At each iteration, we sample a minibatch of transitions from $\mathcal{D}$, draw shared base noises and flow times, generate successor return samples using the target flow, and construct the path-coupled Bellman interpolants. The $\lambda$-target is then formed as a control-variate correction of the sample-based Bellman velocity target. Pseudocode is provided in Appendix A.

# 5. Theory: Bellman-Interpolated Flows and $\lambda$-Targets

## 5.1. Coupled Bellman Interpolants

**Setup.** Fix a policy $\pi$ and a state–action pair $(s, a)$. Let $(S', R) \sim p(\cdot, \cdot \mid s, a)$ and $A' \sim \pi(\cdot \mid S')$. Let $X'_1 \sim \eta_{S', A'}$ denote the terminal next-return (under the true return law), and let $X_0 \sim \mathcal{N}(0, 1)$ denote base noise, independent of the MDP randomness.

**Definition 5.1** (Coupled Bellman interpolants). For $t \in [0, 1)$ define the successor and current interpolants

$$X'_t := t X'_1 + (1 - t) X_0, \tag{14}$$
$$X_t := t (R + \gamma X'_1) + (1 - t) X_0. \tag{15}$$

## 5.2. Bellman-Interpolated Marginals and Posterior Operator

For the independent-source Gaussian special case, define the implied noise (inverse map) for fixed $(x'_1, r, t)$:

$$u(x, x'_1, r, t) := \frac{x - t(r + \gamma x'_1)}{1 - t}. \tag{16}$$

**Definition 5.2** (Bellman-interpolated marginal law). Let

$$P_{s,a,t} := \mathcal{L}((1 - t)X_0 + t(R + \gamma X'_1) \mid S = s, A = a),$$

for $t \in [0, 1)$, under Definition 5.1, where $(X_0, X'_1)$ may be coupled. If $P_{s,a,t}$ is absolutely continuous, write its density as $P_{s,a}(x, t)$. In the special case $X_0 \perp (R, X'_1)$ with $X_0 \sim \mathcal{N}(0, 1)$, this density has the explicit Gaussian-kernel form

$$P_{s,a}(x, t) = \mathbb{E}\left[ \frac{1}{1 - t} \phi(u(x, X'_1, R, t)) \,\Big|\, S = s, A = a \right]. \tag{17}$$

**Proposition 5.3** (Endpoints of the Bellman-interpolated marginals). *Under Definition 5.1 and 5.2:*

1. *At $t = 0$: $X_0$ is the base noise, so $P_{s,a,0} = \mathcal{L}(X_0 \mid S = s, A = a) = \mathcal{N}(0, 1)$.*

2. *As $t \uparrow 1$: $X_t \Rightarrow R + \gamma X'_1$ in distribution, and $P_{s,a,t} \Rightarrow \mathcal{L}(R + \gamma X'_1 \mid S = s, A = a)$ weakly.*

*Defining the endpoint law by*

$$P_{s,a,1} := \mathcal{L}(R + \gamma X'_1 \mid S = s, A = a),$$

*we have $P_{s,a,1} = (T^\pi \eta)_{s,a}$.*

*Proof.* See Appendix C. □

**Definition 5.4** (Posterior operator). For any integrable random variable $g = g(S', A', R, X'_1, X_0, t)$ define

$$\mathcal{B}_{s,a}[g](x, t) := \mathbb{E}\big[g \mid X_t = x, \ S = s, \ A = a\big]. \tag{18}$$

In the independent-source Gaussian case, $\mathcal{B}_{s,a}$ admits the usual Bayes form obtained by weighting samples with the Gaussian kernel in Eq. (17); details are in Appendix C.

## 5.3. Posterior Velocity Identity

**Theorem 5.5** (Continuity equation and posterior velocity). *For each $(s, a)$ and $t \in [0, 1)$, the pathwise velocity is $(R + \gamma X'_1) - X_0$, and the population velocity is*

$$v^\star_{s,a}(x, t) = \mathbb{E}[(R + \gamma X'_1) - X_0 \mid X_t = x, \ S = s, \ A = a].$$

*When $P_{s,a,t}$ has density $P_{s,a}(x, t)$, it satisfies the continuity equation*

$$\partial_t P_{s,a}(x, t) + \partial_x\big(P_{s,a}(x, t)\, v^\star_{s,a}(x, t)\big) = 0,$$

*and equivalently*

$$v^\star_{s,a}(x, t) = \mathcal{B}_{s,a}[(R + \gamma X'_1) - X_0](x, t). \tag{19}$$

**Proof** See Appendix C.

**Connection to flow matching.** Theorem 5.5 identifies the population-optimal velocity field transporting the Bellman-interpolated marginals. PCBF's training objective is precisely a Monte Carlo regression estimator of this conditional expectation; the following analysis shows how the $\lambda$-target changes the estimator's bias/variance.

## 5.4. What the $\lambda$-Target Learns

Let $\bar{v}(\cdot, t, S', A')$ be any (possibly lagged) successor velocity field and define the control variate residual

$$C_{\bar{v}} := \bar{v}(X'_t, t, S', A') - (X'_1 - X_0).$$

Define the per-sample $\lambda$-target

$$v_\lambda := (R + \gamma X'_1) - X_0 + \lambda C_{\bar{v}}. \tag{20}$$

**Proposition 5.6** ($L_2$ regression target and bias decomposition). *Fix $(s, a, t)$. The population minimizer of $\mathbb{E}[(f(X_t) - v_\lambda)^2 \mid S = s, A = a]$ satisfies*

$$f^\star(x) = \mathbb{E}[v_\lambda \mid X_t = x, S = s, A = a] = \mathcal{B}_{s,a}[v_\lambda](x, t).$$

*Consequently the learned marginal field equals*

$$v^{(\lambda)}_{s,a}(x, t) = v^\star_{s,a}(x, t) + \lambda \mathcal{B}_{s,a}[C_{\bar{v}}](x, t). \tag{21}$$

For $\bar{v} = \bar{v}^\star$, the control variate $C_{\bar{v}}$ has mean zero conditional on $(X'_t, S', A')$; however the PCBF regression conditions on the *current* interpolant $X_t$, which generally introduces the bias term in (21). The $\lambda$-scheme matches the population Bellman-interpolant field $v^\star_{s,a}$ iff $\mathcal{B}_{s,a}[C_{\bar{v}}](x, t) = 0$ holds $P_{s,a}(\cdot, t)$-a.e.

### 5.5. Bias of the $\lambda$-Target

The $\lambda$-target introduces a control-variate correction using the successor velocity field. This correction reduces variance but can introduce bias because $C_{\bar{v}}$ is mean-zero when conditioning on the *successor* interpolant $X'_t$, whereas the regression conditions on the *current* interpolant $X_t$. The following bound holds without Gaussianity or deterministic rewards; its proof is in Appendix C.6.

**Proposition 5.7** (Distribution-free $L_2$ bias bound). *Fix $(s, a, t)$ and let $\bar{v}$ be any successor velocity field. Let $\varepsilon_{\mathrm{app}}(s, a, t)$ be the RMS error of $\bar{v}$ relative to the population-optimal successor velocity $\bar{v}^\star$, and let $\varepsilon_{\mathrm{int}}(s, a, t)$ be the intrinsic cross-conditioning bias (both defined precisely in Appendix C.6). Then*

$$\left\| v^{(\lambda)}_{s,a}(\cdot, t) - v^\star_{s,a}(\cdot, t) \right\|_{L_2(P_{s,a,t})} \tag{22}$$
$$\leq |\lambda| \big( \varepsilon_{\mathrm{app}}(s, a, t) + \varepsilon_{\mathrm{int}}(s, a, t) \big).$$

**Gaussian closed form and shared-noise intuition.** To interpret $\varepsilon_{\mathrm{int}}$, consider a one-step linear–Gaussian MRP with deterministic reward $r$, successor return $X'_1 \sim \mathcal{N}(\mu, \sigma^2)$, and base-noise correlation $\rho = \mathrm{Corr}(X_0, X'_0)$. For the population-optimal successor field, the intrinsic residual $C := \bar{v}^\star(Z'_t, t) - (Z'_1 - X'_0)$ satisfies

$$\mathbb{E}[C \mid Z_t = x] = \kappa(t, \gamma, \sigma, \rho)\big(x - t(r + \gamma\mu)\big),$$

$$\kappa(t, \gamma, \sigma, \rho) = \frac{t(1-t)\sigma^2(\rho - \gamma)}{(t^2\sigma^2 + (1-t)^2)((\gamma t)^2\sigma^2 + (1-t)^2)}.$$

Shared-noise coupling ($\rho = 1$) gives $\kappa = O\big((1-\gamma)(1-t)\big)$, so bias vanishes near $\gamma = 1$ and near the flow endpoints; independent paths ($\rho = 0$) retain an $O(\gamma)$ factor at fixed interior $t$. The variance-minimizing coefficient is $\lambda^\star(t) = \gamma(1-t) + \rho t$. Full derivation is in Appendix C.9.

### 5.6. Shared-Noise Bellman Contraction

Let $p \geq 1$. Let $G = \{G_{s,a}\}$ and $H = \{H_{s,a}\}$ be return generators driven by a shared latent seed $\xi$, and define

$$D_p(G, H) := \sup_{s,a} \big( \mathbb{E}_\xi \left[ |G_{s,a}(\xi) - H_{s,a}(\xi)|^p \right] \big)^{1/p}.$$

For a fixed policy $\pi$, define the shared-noise Bellman generator update by

$$(\mathcal{T}_{\mathrm{pc}}G)_{s,a} = R + \gamma G_{S',A'}(\xi'), \quad (S', A') \sim p(\cdot \mid s, a)\pi(\cdot \mid S').$$

**Proposition 5.8** (Shared-noise Bellman contraction). *Under the common coupling where the same transition, action, reward, and successor latent seed $\xi'$ are used when comparing $\mathcal{T}_{\mathrm{pc}}G$ and $\mathcal{T}_{\mathrm{pc}}H$,*

$$D_p(\mathcal{T}_{\mathrm{pc}}G, \mathcal{T}_{\mathrm{pc}}H) \leq \gamma D_p(G, H).$$

*Moreover, for PCBF interpolants $X^\Phi_t = (1 - t)X_0 + t(R + \gamma\Phi_{S',A'}(\xi'))$ with $\Phi \in \{G, H\}$, we have*

$$\sup_{s,a} \big( \mathbb{E}|X^G_t - X^H_t|^p \big)^{1/p} \leq t\gamma D_p(G, H).$$

Proof in Appendix C.4. This is a contraction for the shared coupling used by PCBF. It does not claim contraction of arbitrary independently coupled samples, nor does it by itself give a finite-sample neural optimization guarantee. PCBF can be viewed as a synchronous-coupling method: current and successor return flows are driven by shared latent noise realization, so Bellman comparisons are performed pathwise on aligned trajectories rather than between independently sampled return realizations. This synchronization underlies PCBF's variance reduction, contraction behavior, and improved interpolation stability; the additional $t$-factor indicates that discrepancies vanish near the source distribution and grow gradually over flow time.

## 6. Experiments

In this section, we empirically evaluate PCBF across a diverse set of tasks, ranging from simulated toy environments to large-scale, challenging reinforcement learning benchmarks. We compare PCBF against prior baseline models and further present ablation studies and detailed analyses.

### 6.1. Experimental Setup

**Toy Environments.** To rigorously validate the distributional accuracy of PCBF in a controlled setting, we first consider a suite of analytically tractable toy environments with known return structures. Unlike complex control benchmarks, these environments admit closed-form return laws, allowing us to unambiguously measure the discrepancy between the learned and ground-truth distributions. We utilize three specific environments: (i) **Solitaire Dice**; (ii) **Bernoulli MRP**; and (iii) **Discrete Monte Carlo Chain**. Detailed definitions and additional results for these environments are provided in Appendix I.

**Benchmarks.** We evaluate PCBF on 38 offline RL tasks spanning state-based control, pixel-based visual control, and dexterous manipulation. The primary benchmark is the reward-based single-task variant of OGBench (Park et al., 2025a): four state-based domains (cube-double-play, scene-play, puzzle-4x4-play, cube-triple-play) and two pixel-based

domains (visual-antmaze-teleport, visual-cube-double-play) using $64 \times 64 \times 3$ observations, each with five tasks (30 total). We add eight D4RL Adroit tasks (Fu et al., 2020) (pen, door, hammer,relocate; cloned and expert), covering contact-rich dexterous manipulation. Together these span long-horizon reasoning, compositional manipulation, and semi-sparse rewards. We report mean and standard deviation over eight seeds (four for visual tasks); full environment, implementation, and hyperparameter details are in Appendices H.1, D, and H.2.

**Methods and Evaluation.** We compare PCBF against representative offline RL baselines covering both distributional return modeling and flow-based value learning. For distributional RL baselines, we include **IQN** (Dabney et al., 2018a) and **CODAC** (Ma et al., 2021), which represent return distributions through quantile-based critics. We also compare against **Value Flows** (Dong et al., 2026), a closely related flow-based distributional RL method that directly models the full return distribution using flow matching.

To further evaluate the role of flow-based critic parameterization, we compare with **FloQ** (Agrawalla et al., 2026), which uses flow matching to parameterize scalar Q-functions via iterative numerical integration rather than modeling the full return distribution. In addition, we include strong scalar value-based offline RL baselines, including **IQL** (Kostrikov et al., 2022) and **FQL** (Park et al., 2025b). This comparison allows us to assess whether PCBF's gains arise from flow-based critic parameterization alone or from modeling full return laws with source-consistent Bellman-coupled flows.

### 6.2. Results and Discussion

**OGBench and D4RL.** We evaluate PCBF against a range of strong offline RL baselines on both state-based and pixel-based benchmarks. Table 1 summarizes the aggregated results across 38 tasks, with full per-task scores provided in Table 3 in Appendix G. Overall, PCBF shows a selective rather than uniform advantage over existing offline RL baselines. It achieves the best or near-best aggregate performance on **cube-double-play**, **puzzle-4x4-play**, D4RL Adroit, and **visual-antmaze-teleport**, suggesting that path-coupled Bellman supervision is most beneficial when critic-side return-law fidelity and variance-controlled bootstrapping affect action ranking. On **scene-play**, PCBF remains competitive but trails the strongest flow-based baselines, while on **cube-triple-play** and **visual-cube-double-play** it underperforms Value Flows by a clear margin. These results indicate that improved distributional fitting alone does not resolve all challenges in long-horizon sparse-reward or pixel-based settings, where policy extraction, action-proposal coverage, visual representation, or suboptimal $\lambda$ selection may become the limiting factors.

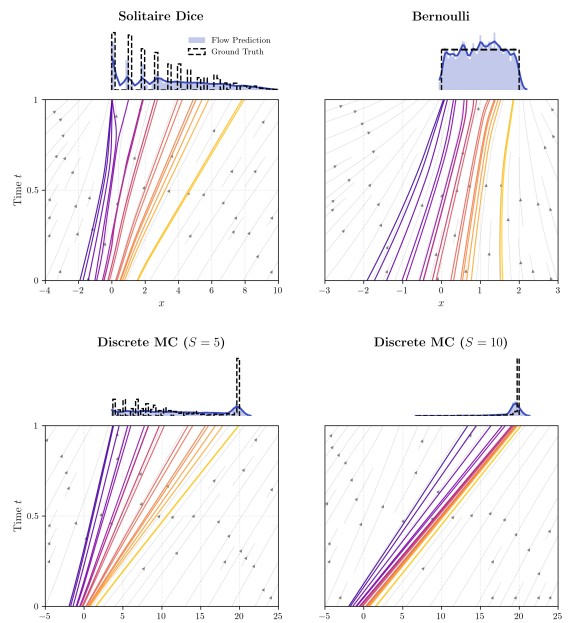

*Figure 2.* **Learned PCBF Maps on Toy Environments.** Left Top (Solitaire); Right Top (Bernoulli); and Bottom (Discrete MC).

**Toy Environments.** To validate the distributional fidelity of PCBF, we visualize the learned probability flows on toy environments with known ground-truth return laws. Figure 2 shows particle trajectories from the Gaussian source distribution at $t = 0$ to the learned return distribution at $t = 1$ for Solitaire Dice, Bernoulli, and Discrete Monte Carlo Chain environments. Across these settings, PCBF recovers the return laws: it captures the heavy-tailed discrete structure in Solitaire Dice, expands source noise over the uniform Bernoulli return support, and reconstructs the multi-modal long-horizon distributions in the Monte Carlo Chain. In all cases, the learned density closely matches the ground-truth histogram or analytic distribution. Additional state-wise flow visualizations for the Monte Carlo Chain are provided in Figure 10 in Appendix I.4.

Additionally, to rigorously assess distributional fidelity, we evaluate PCBF against Value Flows with varying dcfm coefficients on the toy environments. Figure 3 directly compares the learned return CDFs against exact Monte Carlo (MC) references, enabling precise evaluation beyond policy-level metrics. Across all environments, PCBF consistently matches the ground-truth CDFs, while Value Flows (VF) exhibits progressive distributional degradation as dcfm increases—with higher dcfm values causing systematic underestimation of return variance, particularly in long-horizon settings. Extended comparisons across additional states are provided in Appendix I.3 in Figure 9.

Figure 4 contrasts the stability of our method against Value Flows on the Solitaire and Discrete MC tasks. We observe

*Table 1.* **Offline RL Results.** We report performance on OGBench and D4RL. Results are averaged over 8 random seeds (4 seeds for pixel-based tasks). Bold numbers denote the values within 95% of the best performing method on each domain.

| Domain | IQN | CODAC | FloQ | FQL | IQL | Value Flows | PCBF (Ours) |
|---|---|---|---|---|---|---|---|
| cube-double-play (5 tasks) | $42 \pm 8$ | $61 \pm 6$ | $47 \pm 14$ | $29 \pm 6$ | $7 \pm 1$ | $69 \pm 4$ | $\mathbf{71 \pm 5}$ |
| scene-play (5 tasks) | $40 \pm 1$ | $55 \pm 1$ | $\mathbf{58 \pm 4}$ | $56 \pm 2$ | $28 \pm 3$ | $\mathbf{59 \pm 4}$ | $54 \pm 4$ |
| puzzle-4x4-play (5 tasks) | $27 \pm 4$ | $20 \pm 18$ | $28 \pm 6$ | $17 \pm 5$ | $7 \pm 2$ | $27 \pm 4$ | $\mathbf{30 \pm 4}$ |
| cube-triple-play (5 tasks) | $6 \pm 0$ | $2 \pm 1$ | $8 \pm 3$ | $4 \pm 2$ | $1 \pm 1$ | $\mathbf{14 \pm 3}$ | $4 \pm 1$ |
| D4RL adroit (8 tasks) | $66 \pm 5$ | $69 \pm 0$ | $70 \pm 5$ | $\mathbf{71 \pm 4}$ | $70$ | $65 \pm 2$ | $\mathbf{69 \pm 2}$ |
| visual-antmaze-teleport (5 tasks) | $4 \pm 2$ | – | – | $5 \pm 2$ | $6 \pm 4$ | $13 \pm 4$ | $\mathbf{14 \pm 4}$ |
| visual-cube-double-play (5 tasks) | $1 \pm 0$ | – | – | $6 \pm 1$ | $11 \pm 6$ | $\mathbf{13 \pm 2}$ | $3 \pm 0$ |

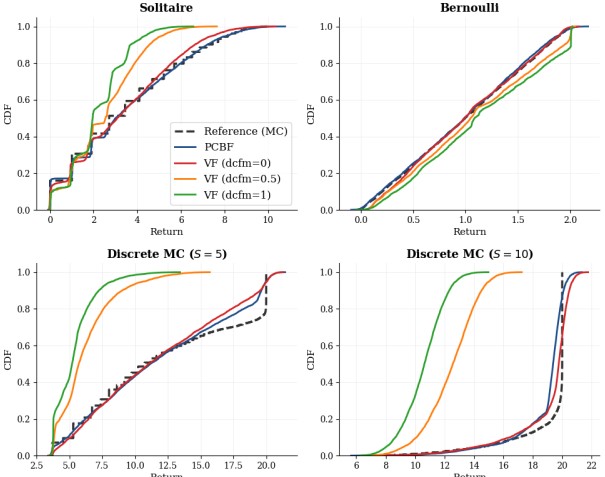

*Figure 3.* **Distributional accuracy comparison on toy environments.** Learned return CDFs for PCBF and Value Flows (with dcfm $\in \{0, 0.5, 1\}$) compared against ground-truth references.

that increasing the consistency coefficient (`dcfm`) in Value Flows systematically degrades distributional accuracy, suggesting that strict trajectory-wide consistency conflicts with the boundary conditions required for accurate optimal transport. In contrast, PCBF's $\lambda$-target effectively decouples variance reduction from bias, consistently outperforming Value Flows across nearly all hyperparameter settings in both environments while maintaining low Wasserstein error.

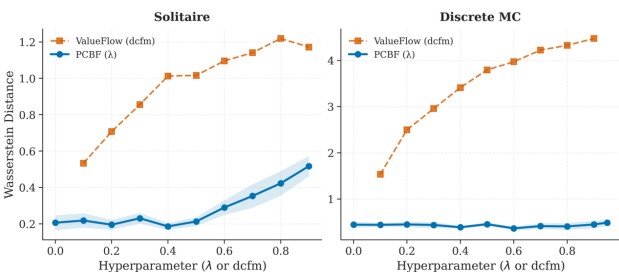

*Figure 4.* **Hyperparameter Sensitivity Analysis (PCBF vs. Value Flows) on Solitaire and Discrete MC Environments.** A more detailed discussion can be found in Figure 8 of Appendix I.3.

## 7. Limitations

Despite its advantages, PCBF exhibits two primary limitations. **(i)** Computational cost: training and evaluation require numerical integration of a velocity field; Table 2 shows PCBF is in the same ballpark as other flow-based critics but slower than scalar methods. **(ii)** $\lambda$ is task-dependent: we tune it once per domain, while fully adaptive online selection is left for future work.

## 8. Conclusion

We presented Path-Coupled Bellman Flows, a continuous-time distributional RL framework whose *source-consistent Bellman-coupled paths* and shared-noise coupling make flow matching compatible with the Bellman endpoint while controlling target variance, with a $\lambda$-parameterized control variate trading off bias and variance. We supported it with population-velocity identification, an $L_2$ bias analysis, shared-noise contraction, and an Euler-sensitivity bound. Empirically, PCBF recovers known return laws on toy MRPs, achieves strong results on several OGBench domains and competitive D4RL performance, and shows smaller corrected pathwise Bellman residuals under coarse discretization when coupling is enabled.

## Acknowledgements

The authors thank Arizona State University Research Computing for A100 GPU resources on the Sol supercomputer (Jennewein et al., 2023).

## Impact Statement

This paper presents methodological advancements in DRL. By improving the return distribution accuracy and stability, our work enables safer, risk-sensitive decision-making in uncertain environments like robotics or autonomous control. While advancing autonomous systems carries general societal implications regarding automation and safety, this work focuses on fundamental algorithmic improvements and does not introduce ethical or privacy concerns.

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

# A. PCBF Training Algorithm

---

**Algorithm 1** PCBF Training (offline RL)

---

**Input:** offline dataset $\mathcal{D}$, discount $\gamma \in (0, 1)$, batch size $B$, control parameter $\lambda$, target update rate $\tau$
Define $\text{sg}(\cdot)$ as stop-gradient (no backprop through the argument).
Initialize current velocity network $v_\theta$
Initialize target network $v_{\theta^-} \leftarrow v_\theta$
**repeat**
   Sample minibatch $\{(S_i, A_i, R_i, S'_i, d_i)\}_{i=1}^B \sim \mathcal{D}$
   **for** $i = 1$ **to** $B$ **do**
      Sample next action $A'_i \sim \pi(\cdot | S'_i)$
      Sample $X_{0,i} \sim \mathcal{N}(0, 1)$ and $t_i \sim \text{Unif}[0, 1]$
      $X'_i \leftarrow \psi^1_{\theta^-}(X_{0,i} \mid S'_i, A'_i)$
      $\tilde{\gamma}_i \leftarrow \gamma(1 - d_i)$ {effective discount; 0 if terminal}
      $\lambda_i \leftarrow (1 - d_i)\lambda$ {no successor correction at terminal transitions}
      $Z^{s'}_{t,i} \leftarrow (1 - t_i)X_{0,i} + t_i X'_i$
      $Z^s_{t,i} \leftarrow t_i R_i + \tilde{\gamma}_i Z^{s'}_{t,i} + (1 - t_i)(1 - \tilde{\gamma}_i)X_{0,i}$
      $c_i \leftarrow v_{\theta^-}(t_i, Z^{s'}_{t,i} \mid S'_i, A'_i)$
      $C_i \leftarrow c_i - (X'_i - X_{0,i})$
      $Y_i \leftarrow R_i + \tilde{\gamma}_i X'_i - X_{0,i}$
      $u_i \leftarrow Y_i + \lambda_i C_i$
   **end for**
   $\mathcal{L} \leftarrow \frac{1}{B}\sum_{i=1}^B \|v_\theta(t_i, Z^s_{t,i} \mid S_i, A_i) - \text{sg}(u_i)\|_2^2$
   Take a gradient step on $\theta$ to minimize $\mathcal{L}$
   $\theta^- \leftarrow \tau\theta + (1 - \tau)\theta^-$
**until** convergence

---

# B. Value Flows (DCFM/BCFM) and Comparison with PCBF

This appendix complements Lemma 4.1: we summarize Value Flows (Dong et al., 2026), state its DCFM/BCFM training objectives, explain the source-boundary tension under a literal full-$t$ distributional Bellman constraint at $t=0$, and contrast this with PCBF's coupled-path construction.

Value Flows learns a time-dependent vector field $v : \mathbb{R} \times [0, 1] \times \mathcal{S} \times \mathcal{A} \to \mathbb{R}$ that generates a diffeomorphic flow $\psi$ transforming samples from a standard Gaussian $X_0 \sim \mathcal{N}(0, 1)$ into return samples. The flow satisfies the ODE $\frac{d}{dt}\psi(X_0|t, s, a) = v(\psi(X_0|t, s, a)|t, s, a)$ with $\psi(X_0|0, s, a) = X_0$.

To train the return vector field, Value Flows combines two losses. The first is the Distributional Conditional Flow Matching (DCFM) loss, which matches the velocity field recursively through bootstrapping from a target network. The second is the Bootstrapped Conditional Flow Matching (BCFM) loss, which provides direct supervision using TD targets and is added as a stabilizing regularizer:

$$\mathcal{L}_{\text{BCFM}}(v) = \mathbb{E}_{\mathcal{D}}\left[\left\|v(t, Z^{\text{TD}}_t|s, a) - (Z^{\text{TD}}_1 - X_0)\right\|^2\right] \tag{23}$$

where $Z^{\text{TD}}_1 = R(s, a) + \gamma X'$ is the bootstrapped return (with $X'$ sampled from the target return distribution at $s', a'$), and $Z^{\text{TD}}_t = (1 - t)X_0 + tZ^{\text{TD}}_1$ is the linear interpolation between the noise and the target return. The final loss is $\mathcal{L}_{\text{Value Flows}} = \mathcal{L}_{\text{DCFM}} + \lambda\mathcal{L}_{\text{BCFM}}$. This $\lambda$ in Value Flows is a loss-balancing coefficient and should not be confused with the PCBF control-variate parameter $\lambda$.

**Remark on DCFM scaling.** In the Value Flows paper, the practical DCFM loss is implemented without an explicit $\gamma$ multiplier on the successor velocity,

$$\mathcal{L}^{\text{VF}}_{\text{DCFM}}(v, v_k) = \mathbb{E}\left[\|v(t, Z_t \mid s, a) - v_k(t, Z'_t \mid s', a')\|^2\right], \qquad Z'_t = \frac{Z_t - R}{\gamma},$$

with BCFM added as a stabilizing regularizer; the authors also report that naive DCFM alone produced a divergent vector field. We record the geometry-consistent corrected analogue implied by the affine Bellman path $Z_t = R + \gamma Z_t'$, whose time derivative gives $\dot{Z}_t = \gamma \dot{Z}_t'$:

$$\mathcal{L}_{\mathrm{DCFM}}(v, v_k) = \mathbb{E}\left[\left\|v(t, Z_t \mid s, a) - \gamma\, v_k\left(t, \frac{Z_t - R}{\gamma} \,\Big|\, s', a'\right)\right\|^2\right].$$

These are different functional equations: the unscaled relation $v(R + \gamma z, t) = v(z, t)$ and the scaled relation $v(R + \gamma z, t) = \gamma v(z, t)$ admit different solution classes (e.g., under $R = 0$, the former can force $v(\gamma z, t) = v(z, t)$ and hence constant-in-$z$ behavior, whereas the latter permits homogeneous/linear solutions). Our degeneracy analysis below applies only to the original unscaled Value-Flows-style DCFM condition.

### B.1. Bellman Inconsistency in Value Flows

**Direct fixed-point incompatibility at $t = 0$ (distribution-level statement).** In this 1-state MRP, "Bellman at each $t$" implies that the time-$t$ marginal density must satisfy

$$p_v(\cdot \mid t) \overset{!}{=} \mathcal{T} p_v(\cdot \mid t) := \mathbb{E}_R\left[\frac{1}{\gamma}p_v\left(\frac{\cdot - R}{\gamma} \,\Big|\, t\right)\right],$$

which is exactly the density form of the distributional Bellman equation.

Because $\mathcal{T}$ has a unique fixed point distribution $p_Z^\star$ (the return law), the condition above implies $p_v(\cdot \mid t) = p_Z^\star$ for every $t$. In particular, it would require $p_v(\cdot \mid 0) = p_Z^\star$. However, the flow boundary condition enforces $p_v(\cdot \mid 0) = \mathcal{N}(0, 1)$.

Consequently, a literal full-$t$ distributional Bellman fixed-point constraint would be incompatible with the Gaussian source boundary. A literal full-$t$ DCFM-style Bellman density constraint inherits this source-boundary tension; the practical Value Flows objective mitigates it by adding BCFM anchoring and uncertainty weighting, consistent with the authors' ablations showing that BCFM regularization is important.

We formalize the resulting degeneracy (constant-in-$z$ solutions under vanishing unscaled DCFM loss) as follows. We do not claim that Value Flows as a whole is invalid; rather, our concern is that strongly overweighting the DCFM term relative to BCFM anchoring can bias the intermediate vector field toward source-incompatible solutions, whereas PCBF separates endpoint Bellman consistency from intermediate path construction. In contrast, PCBF achieves Bellman consistency by explicitly coupling the flow paths so that the terminal distribution at $t = 1$ satisfies the Bellman equation pointwise while respecting the prescribed Gaussian source at $t=0$, avoiding the full-$t$ fixed-point conflict discussed above.

## C. Proof Sketches

### C.1. Derivation of the Density Formula and Proof of Proposition 5.3

Definition 5.2 is stated at the level of the pushforward law $P_{s,a,t} = \mathcal{L}(X_t \mid s, a)$ and does not require independence between $X_0$ and $X_1'$. The explicit kernel formula (17) is the independent-source Gaussian special case obtained by change-of-variables using $u(x, x_1', r, t)$ from (16) and the Jacobian $1/(1 - t)$.

**Proof of Proposition 5.3:**

1. At $t = 0$, both interpolants in (14)–(15) reduce to the base noise $X_0$, hence $P_{s,a,0} = \mathcal{N}(0, 1)$.

2. As $t \to 1$, we have $X_t = t(R + \gamma X_1') + (1 - t)X_0 \to R + \gamma X_1'$ pointwise, hence $X_t \Rightarrow R + \gamma X_1'$ in distribution. The weak convergence of $P_{s,a}(\cdot, t)$ follows, and pointwise convergence of densities holds under dominated convergence when the limit law is absolutely continuous.

### C.2. Properties of the Posterior Operator

The posterior operator $\mathcal{B}_{s,a}$ defined in (18) satisfies:

1. **(Linearity)** $\mathcal{B}_{s,a}[\alpha g_1 + \beta g_2](x, t) = \alpha \mathcal{B}_{s,a}[g_1](x, t) + \beta \mathcal{B}_{s,a}[g_2](x, t)$.

2. (**Tower property**) If $h = h(X_t, S, A)$ is measurable w.r.t. $(X_t, S, A)$, then $\mathcal{B}_{s,a}[h \cdot g](x, t) = h(x, s, a) \, \mathcal{B}_{s,a}[g](x, t)$ and $\mathbb{E}[\mathcal{B}_{s,a}[g](X_t, t) \mid S = s, A = a] = \mathbb{E}[g \mid S = s, A = a]$.

3. (**Bayes form, independent-source special case**) In the independent-source Gaussian case, Bayes' rule yields:

$$P_{s,a}(x, t) = \widetilde{\mathcal{B}}_{s,a}[1](x, t), \tag{24}$$

$$\mathcal{B}_{s,a}[g](x, t) = \frac{\widetilde{\mathcal{B}}_{s,a}[g](x, t)}{\widetilde{\mathcal{B}}_{s,a}[1](x, t)} = \frac{\widetilde{\mathcal{B}}_{s,a}[g](x, t)}{P_{s,a}(x, t)}. \tag{25}$$

This is the continuous form of Bayes' rule: the density $P_{s,a}$ normalizes the unnormalized posterior to yield the conditional expectation $\mathcal{B}_{s,a}$.

These follow from standard properties of conditional expectation; Eqs. (24)–(25) additionally use the independent-source Gaussian representation (17).

### C.3. Proof of Theorem 5.5

Differentiate $X_t = t(R + \gamma X_1') + (1 - t)X_0$ to obtain the pathwise velocity $(R + \gamma X_1') - X_0$. Taking conditional expectation given $(X_t = x, S = s, A = a)$ yields (19). When $P_{s,a,t}$ is absolutely continuous, the continuity equation follows from the weak form for smooth compactly supported test functions $\varphi$ and under the required integrability assumptions:

$$\frac{d}{dt}\mathbb{E}[\varphi(X_t) \mid s, a] = \mathbb{E}[\varphi'(X_t)\left((R + \gamma X_1') - X_0\right) \mid s, a]$$

and disintegration with respect to $X_t$.

### C.4. Proof of Proposition 5.8

For any $(s, a)$, using the common coupling cancels the reward:

$$(\mathcal{T}_{\mathrm{pc}}G)_{s,a} - (\mathcal{T}_{\mathrm{pc}}H)_{s,a} = \gamma\big(G_{S',A'}(\xi') - H_{S',A'}(\xi')\big).$$

Thus

$$\left(\mathbb{E}\left[|(\mathcal{T}_{\mathrm{pc}}G)_{s,a} - (\mathcal{T}_{\mathrm{pc}}H)_{s,a}|^p\right]\right)^{1/p} = \gamma\left(\mathbb{E}\left[|G_{S',A'}(\xi') - H_{S',A'}(\xi')|^p\right]\right)^{1/p} \leq \gamma D_p(G, H).$$

Taking the supremum over $(s, a)$ gives the first claim. For the interpolants,

$$X_t^G - X_t^H = t\gamma\big(G_{S',A'}(\xi') - H_{S',A'}(\xi')\big),$$

so the same argument gives the $t\gamma$ bound.

### C.5. Proof of Proposition 5.6

Use the standard $L_2$ projection lemma: the minimizer of $\mathbb{E}[(f(X) - Y)^2]$ is $f(x) = \mathbb{E}[Y \mid X = x]$. Apply with $X = X_t$ and $Y = v_\lambda$ to get $f^\star(x) = \mathcal{B}_{s,a}[v_\lambda](x, t)$, then expand $v_\lambda$ using (20) to get (21).

### C.6. Proof of Proposition 5.7 (general bound)

*Why this matters:* the bound controls $\lambda$-induced population bias in the same $L_2$ geometry as the velocity regression, without assuming Gaussianity, unimodality, or deterministic rewards.

**Notation.** Recall the control-variate residual

$$C_{\bar{v}} := \bar{v}(X_t', t, S', A') - (X_1' - X_0),$$

the population-optimal successor velocity

$$\bar{v}^\star(z', t, s', a') := \mathbb{E}[X_1' - X_0 \mid X_t' = z', S' = s', A' = a'],$$

and the decomposition $C_{\bar{v}} = E_{\text{app}} + E_{\text{int}}$ with

$$E_{\text{app}} := \bar{v}(X'_t, t, S', A') - \bar{v}^{\star}(X'_t, t, S', A'), \qquad E_{\text{int}} := \bar{v}^{\star}(X'_t, t, S', A') - (X'_1 - X_0).$$

For $h \in L_2(P_{s,a,t})$, write $\|h\|^2_{L_2(P_{s,a,t})} := \mathbb{E}[h(X_t)^2 \mid S = s, A = a]$, and define

$$\varepsilon_{\text{app}}(s, a, t) := \left(\mathbb{E}\big[|E_{\text{app}}|^2 \big| S = s, A = a\big]\right)^{1/2}, \qquad \varepsilon_{\text{int}}(s, a, t) := \|\mathcal{B}_{s,a}[E_{\text{int}}](\cdot, t)\|_{L_2(P_{s,a,t})}.$$

**Proof.** We prove the integrated $L_2$ bias bound for the $\lambda$-target. Recall that

$$C_{\bar{v}} = \bar{v}(X'_t, t, S', A') - (X'_1 - X_0).$$

Add and subtract the population-optimal successor velocity

$$\bar{v}^{\star}(X'_t, t, S', A') = \mathbb{E}[X'_1 - X_0 \mid X'_t, S', A']$$

to decompose

$$C_{\bar{v}} = E_{\text{app}} + E_{\text{int}},$$

where

$$E_{\text{app}} := \bar{v}(X'_t, t, S', A') - \bar{v}^{\star}(X'_t, t, S', A')$$

and

$$E_{\text{int}} := \bar{v}^{\star}(X'_t, t, S', A') - (X'_1 - X_0).$$

By construction,

$$\mathbb{E}[E_{\text{int}} \mid X'_t, S', A'] = 0. \tag{26}$$

However, the PCBF regression conditions on the current interpolant $X_t$, not on the successor interpolant $X'_t$, so generally

$$\mathbb{E}[E_{\text{int}} \mid X_t, S = s, A = a] \neq 0.$$

This is the source of the $\lambda$-induced intrinsic bias.

From Proposition 5.6,

$$v^{(\lambda)}_{s,a}(x, t) - v^{\star}_{s,a}(x, t) = \lambda \, \mathcal{B}_{s,a}[C_{\bar{v}}](x, t).$$

Therefore

$$\left\|v^{(\lambda)}_{s,a}(\cdot, t) - v^{\star}_{s,a}(\cdot, t)\right\|_{L_2(P_{s,a,t})} = |\lambda| \, \|\mathcal{B}_{s,a}[C_{\bar{v}}](\cdot, t)\|_{L_2(P_{s,a,t})}.$$

Using $C_{\bar{v}} = E_{\text{app}} + E_{\text{int}}$ and the triangle inequality in $L_2(P_{s,a,t})$,

$$\|\mathcal{B}_{s,a}[C_{\bar{v}}]\|_{L_2(P_{s,a,t})} \leq \|\mathcal{B}_{s,a}[E_{\text{app}}]\|_{L_2(P_{s,a,t})} + \|\mathcal{B}_{s,a}[E_{\text{int}}]\|_{L_2(P_{s,a,t})}. \tag{27}$$

For the approximation term, conditional Jensen gives

$$|\mathcal{B}_{s,a}[E_{\text{app}}](X_t, t)|^2 = |\mathbb{E}[E_{\text{app}} \mid X_t, S = s, A = a]|^2 \leq \mathbb{E}[|E_{\text{app}}|^2 \mid X_t, S = s, A = a].$$

Taking expectation over $X_t \sim P_{s,a,t}$ and applying the tower property yields

$$\|\mathcal{B}_{s,a}[E_{\text{app}}]\|^2_{L_2(P_{s,a,t})} \leq \mathbb{E}[|E_{\text{app}}|^2 \mid S = s, A = a] \tag{28}$$

$$= \mathbb{E}\Big[\big|\bar{v}(X'_t, t, S', A') - \bar{v}^{\star}(X'_t, t, S', A')\big|^2 \Big| S = s, A = a\Big]. \tag{29}$$

Thus

$$\|\mathcal{B}_{s,a}[E_{\text{app}}]\|_{L_2(P_{s,a,t})} \leq \varepsilon_{\text{app}}(s, a, t).$$

For the intrinsic term, by definition

$$\varepsilon_{\text{int}}(s, a, t) = \|\mathcal{B}_{s,a}[E_{\text{int}}]\|_{L_2(P_{s,a,t})}.$$

Combining this identity with (27) gives

$$\|\mathcal{B}_{s,a}[C_{\bar{v}}]\|_{L_2(P_{s,a,t})} \leq \varepsilon_{\mathrm{app}}(s,a,t) + \varepsilon_{\mathrm{int}}(s,a,t).$$

Multiplying by $|\lambda|$ proves (22).

It remains to prove the auxiliary upper bound on $\varepsilon_{\mathrm{int}}$. Again by conditional Jensen,

$$|\mathbb{E}[E_{\mathrm{int}} \mid X_t, S = s, A = a]|^2 \leq \mathbb{E}[|E_{\mathrm{int}}|^2 \mid X_t, S = s, A = a].$$

Taking expectation over $X_t$ and applying the tower property gives

$$\varepsilon_{\mathrm{int}}(s,a,t)^2 \leq \mathbb{E}[|E_{\mathrm{int}}|^2 \mid S = s, A = a].$$

Moreover,

$$\varepsilon_{\mathrm{int}}(s,a,t) \leq \left(\mathbb{E}\big[|E_{\mathrm{int}}|^2 \big| S = s, A = a\big]\right)^{1/2}. \tag{30}$$

**Interpretation.** The bound separates $\lambda$-bias into the successor-flow approximation term $\varepsilon_{\mathrm{app}}$ and the cross-conditioning intrinsic term $\varepsilon_{\mathrm{int}}$, which arises because PCBF regresses on the current interpolant $X_t$ rather than on $(X_t', S', A')$, under which the control variate is mean-zero. Thus $\lambda > 0$ is unbiased only when $\varepsilon_{\mathrm{int}} = 0$. The bound scales linearly in $|\lambda|$, so $\lambda = 0$ recovers the unbiased sample target while larger $\lambda$ trades bias for variance reduction.

### C.7. Pointwise Refinements for Proposition 5.7

The integrated bound is the main result because PCBF is trained by an $L_2$ velocity regression under the current interpolant marginal. Pointwise bounds are possible, but they must condition on the local posterior law or pay an explicit density-ratio factor.

**Corollary C.1** (Pointwise conditional-law bound). *Let*

$$\mu_{x,t} := \mathcal{L}(X_t', S', A' \mid X_t = x, S = s, A = a)$$

*be the conditional law of the successor-side variables given the current interpolant. For $P_{s,a,t}$-a.e. $x$,*

$$|\mathcal{B}_{s,a}[C_{\bar{v}}](x,t)| \leq \|\bar{v} - \bar{v}^\star\|_{L_2(\mu_{x,t})} + \rho_{\mathrm{int}}(x,t), \tag{31}$$

*where*

$$\rho_{\mathrm{int}}(x,t) := \left(\mathbb{E}[|E_{\mathrm{int}}|^2 \mid X_t = x, S = s, A = a]\right)^{1/2}$$

*is a root conditional second moment. Consequently,*

$$\left|v_{s,a}^{(\lambda)}(x,t) - v_{s,a}^\star(x,t)\right| \leq |\lambda| \left(\|\bar{v} - \bar{v}^\star\|_{L_2(\mu_{x,t})} + \rho_{\mathrm{int}}(x,t)\right).$$

*Proof.* Fix $x$ in the support of $P_{s,a,t}$. Using $C_{\bar{v}} = E_{\mathrm{app}} + E_{\mathrm{int}}$,

$$|\mathcal{B}_{s,a}[C_{\bar{v}}](x,t)| \leq |\mathbb{E}[E_{\mathrm{app}} \mid X_t = x, S = s, A = a]| + |\mathbb{E}[E_{\mathrm{int}} \mid X_t = x, S = s, A = a]|.$$

For the first term, conditional Jensen and the definition of $\mu_{x,t}$ give

$$|\mathbb{E}[E_{\mathrm{app}} \mid X_t = x, S = s, A = a]| \leq \mathbb{E}[|E_{\mathrm{app}}| \mid X_t = x, S = s, A = a] \tag{32}$$
$$\leq \left(\mathbb{E}[|E_{\mathrm{app}}|^2 \mid X_t = x, S = s, A = a]\right)^{1/2} \tag{33}$$
$$= \|\bar{v} - \bar{v}^\star\|_{L_2(\mu_{x,t})}. \tag{34}$$

For the second term, Cauchy–Schwarz gives

$$|\mathbb{E}[E_{\mathrm{int}} \mid X_t = x, S = s, A = a]| \leq \left(\mathbb{E}[|E_{\mathrm{int}}|^2 \mid X_t = x, S = s, A = a]\right)^{1/2} = \rho_{\mathrm{int}}(x,t).$$

Combining the two inequalities proves the claim. $\square$

**Corollary C.2** (Pointwise bound with density-ratio factor). *Let*

$$\nu_t := \mathcal{L}(X'_t, S', A' \mid S = s, A = a)$$

*and suppose that, for $P_{s,a,t}$-a.e. $x$,*

$$\mu_{x,t} := \mathcal{L}(X'_t, S', A' \mid X_t = x, S = s, A = a) \ll \nu_t.$$

*Let*

$$w_{x,t} := \frac{d\mu_{x,t}}{d\nu_t}, \qquad \Omega_{x,t} := \|w_{x,t}\|_{L_2(\nu_t)}.$$

*Then*

$$|\mathcal{B}_{s,a}[C_{\bar{v}}](x,t)| \leq \Omega_{x,t} \|\bar{v} - \bar{v}^\star\|_{L_2(\nu_t)} + \rho_{\text{int}}(x,t), \tag{35}$$

*where*

$$\rho_{\text{int}}(x,t) := \left(\mathbb{E}[|E_{\text{int}}|^2 \mid X_t = x, S = s, A = a]\right)^{1/2}.$$

*Proof.* Let

$$e(z', s', a') := \bar{v}(z', t, s', a') - \bar{v}^\star(z', t, s', a').$$

For $P_{s,a,t}$-a.e. $x$, by absolute continuity,

$$\mathbb{E}[E_{\text{app}} \mid X_t = x, S = s, A = a] = \int e(y) \, d\mu_{x,t}(y) = \int e(y) w_{x,t}(y) \, d\nu_t(y),$$

where $y = (z', s', a')$. By Cauchy–Schwarz under $\nu_t$,

$$|\mathbb{E}[E_{\text{app}} \mid X_t = x, S = s, A = a]| \leq \left(\int |e(y)|^2 \, d\nu_t(y)\right)^{1/2} \left(\int |w_{x,t}(y)|^2 \, d\nu_t(y)\right)^{1/2}.$$

Therefore

$$|\mathbb{E}[E_{\text{app}} \mid X_t = x, S = s, A = a]| \leq \Omega_{x,t} \|\bar{v} - \bar{v}^\star\|_{L_2(\nu_t)}.$$

The intrinsic term is bounded exactly as in Corollary C.1:

$$|\mathbb{E}[E_{\text{int}} \mid X_t = x, S = s, A = a]| \leq \rho_{\text{int}}(x,t).$$

Combining the two terms proves (35). $\square$

### C.8. Euler Integration Error in the $\lambda$-Target

The training target uses a numerically integrated successor endpoint $\hat{X}'$ instead of an ideal $X'$. The following proposition isolates how this perturbation propagates into the $\lambda$-target on *nonterminal* transitions; at terminals $\tilde{\gamma} = 0$ and $\lambda_i = 0$ in Algorithm 1, so successor endpoint error does not enter.

**Proposition C.3** (Endpoint integration error in the $\lambda$-target). *Let $X'$ be the exact successor endpoint under the target flow map and $\hat{X}' = X' + \delta$ the endpoint returned by a numerical ODE solver. Define $Z'_t = (1 - t)X_0 + tX'$ and $\hat{Z}'_t = (1 - t)X_0 + t\hat{X}'$. Assume the successor velocity field $\bar{v}(t, z, s', a')$ is $L_t$-Lipschitz in $z$ for the relevant range. For a nonterminal transition, define*

$$u^\lambda := R + \gamma X' - X_0 + \lambda\big(\bar{v}(t, Z'_t, s', a') - (X' - X_0)\big),$$

*and define $\hat{u}^\lambda$ analogously with $(\hat{X}', \hat{Z}'_t)$. Then*

$$\|\hat{u}^\lambda - u^\lambda\| \leq \big(|\gamma - \lambda| + \lambda L_t t\big)\|\delta\|.$$

*In particular, if $0 \leq \lambda \leq \gamma$,*

$$\|\hat{u}^\lambda - u^\lambda\| \leq \big(\gamma - \lambda + \lambda L_t t\big)\|\delta\|.$$

*Proof.* Write $u^\lambda = (R - X_0) + (\gamma - \lambda)X' + \lambda\bar{v}(t, Z'_t, s', a')$ after expanding the bracket. Then $\hat{u}^\lambda - u^\lambda = (\gamma - \lambda)(\hat{X}' - X') + \lambda(\bar{v}(t, \hat{Z}'_t) - \bar{v}(t, Z'_t))$. Use $\|\hat{X}' - X'\| = \|\delta\|$ and Lipschitz continuity in the second term: $\|\bar{v}(t, \hat{Z}'_t) - \bar{v}(t, Z'_t)\| \leq L_t\|\hat{Z}'_t - Z'_t\| = L_t t\|\delta\|$. Combine terms. $\square$

**Interpretation.** At $\lambda = 0$ the bound scales with $\gamma\|\delta\|$, reflecting usual sensitivity to the bootstrapped endpoint. When $0 \leq \lambda \leq \gamma$, choosing $\lambda$ closer to $\gamma$ replaces part of the explicit $\gamma X'$ dependence with the smoother field $\bar{v}$, reducing the coefficient in front of $\|\delta\|$ whenever $L_t t < 1$ (and potentially increasing it in nonsmooth regimes).

### C.9. Proof of the Gaussian Special Case in Proposition 5.7

We prove the linear–Gaussian instantiation stated in Proposition 5.7. Fix $t \in (0, 1)$, $\gamma \in (0, 1)$, deterministic reward $R = r$, successor return $Z'_1 \sim \mathcal{N}(\mu, \sigma^2)$ with $\sigma > 0$, and a bivariate standard noise pair $(X_0, X'_0)$ (independent of $Z'_1$) with $\text{Corr}(X_0, X'_0) = \rho \in [-1, 1]$. Form $Z'_t := tZ'_1 + (1-t)X'_0$, $Z_t := t(r + \gamma Z'_1) + (1-t)X_0$, let $\bar{v}^\star(\cdot, t)$ be the population successor velocity, and set $C := \bar{v}^\star(Z'_t, t) - (Z'_1 - X'_0)$.

**Derivation.** Write $Z'_1 = \mu + \sigma W$ with $W \sim \mathcal{N}(0, 1)$, and represent $(X_0, X'_0)$ as $X'_0 = V'$, $X_0 = \rho V' + \sqrt{1 - \rho^2}\, V$ with $V, V' \sim \mathcal{N}(0, 1)$ independent, $W \perp (V, V')$. For $Z'_t = t(\mu + \sigma W) + (1-t)V'$, the Gaussian regression formula gives $\bar{v}^\star(z', t) = \mathbb{E}[Z'_1 - X'_0 \mid Z'_t = z'] = \mu + \beta(t, \sigma)(z' - t\mu)$, with $\beta(t, \sigma) = (t\sigma^2 - (1-t))/(t^2\sigma^2 + (1-t)^2)$. Substituting $z' = Z'_t$ and subtracting $Z'_1 - X'_0$ yields the linear form $C = a(t, \sigma)W + b(t, \sigma)V'$ with $a = -\sigma(1-t)/(t^2\sigma^2 + (1-t)^2)$, $b = t\sigma^2/(t^2\sigma^2 + (1-t)^2)$. Meanwhile $Z_t - \mathbb{E}[Z_t] = \gamma t\sigma W + (1-t)(\rho V' + \sqrt{1 - \rho^2}\, V)$, and joint Gaussianity yields $\mathbb{E}[C \mid Z_t = x] = \text{Cov}(C, Z_t)/\text{Var}(Z_t)\,(x - \mathbb{E}[Z_t])$. A direct covariance calculation gives $\text{Cov}(C, Z_t) = t(1-t)\sigma^2(\rho - \gamma)/(t^2\sigma^2 + (1-t)^2)$ and $\text{Var}(Z_t) = (\gamma t)^2\sigma^2 + (1-t)^2$, which yields $\kappa(t, \gamma, \sigma, \rho)$. For the $\lambda$-target $u_t^\lambda := (r + \gamma Z'_1) - X_0 + \lambda C$ and $D_t := t^2\sigma^2 + (1-t)^2$,

$$\text{Var}(u_t^\lambda \mid t) = 1 + \gamma^2\sigma^2 + \frac{\sigma^2}{D_t}(\lambda^2 - 2\lambda(\gamma(1-t) + \rho t)),$$

so $\lambda^\star(t) = \gamma(1-t) + \rho t$ minimizes the target variance.

**Consequence (shared-noise vs. independent paths).** Setting $\rho = 1$ (PCBF's shared noise) gives $\kappa = O((1-\gamma)(1-t)) \to 0$ as either $\gamma \to 1$ or $t \to 1$, so the bias contracts at the endpoints and for semi-sparse-reward, near-undiscounted regimes. Setting $\rho = 0$ (independent noise) leaves an $O(\gamma)$ multiplicative factor that does not vanish, illustrating quantitatively why path coupling is essential for the variance-bias trade-off.

### C.10. PCBF Endpoint Bellman Consistency

For the PCBF path in Eq. (9), the endpoint satisfies

$$Z_1^s = R + \gamma X',$$

pointwise. Therefore, if $X' \sim \eta_{s', a'}$, then

$$\mathcal{L}(Z_1^s \mid s, a) = \mathcal{L}(R + \gamma X' \mid s, a) = (T^\pi\eta)_{s,a},$$

which is exactly the distributional Bellman update at the endpoint.

## D. Implementation Details

**Flow matching.** We implement PCBF using a **path-coupled flow-matching** formulation with linear interpolation paths and uniform time sampling, as described in Sections 3 and 4. The return distribution is modeled via a time-dependent velocity field and trained using a mean squared error objective defined by the path-coupled Bellman targets. We numerically solve the resulting ODE using the explicit Euler method with **10 integration steps** across all experiments. For simplicity and stability, we do not use additional time embeddings for the flow time variable. Ablation results for different values of the $\lambda$ parameter, which controls the strength of the path-coupled correction, are reported in Appendix F.

**Value learning.** PCBF learns return distributions rather than scalar value functions. We maintain a target velocity network, updated using Polyak averaging, to generate successor return samples for the path-coupled Bellman objective. The $\lambda$-parameterized target interpolates between a sample-based Bellman-consistent update and a variance-reduced correction derived from successor flow predictions. All value learning components are trained using squared regression losses on the induced velocity targets.

**Network architectures.** For all state-based tasks, we use multilayer perceptrons with four hidden layers of width 512 and GELU activations to parameterize the velocity fields. Layer normalization is applied to stabilize training. For pixel-based environments, we employ a lightweight convolutional encoder adapted from the IMPALA architecture to extract visual features before feeding them into the flow model.

**Image processing.** For pixel-based environments, observations consist of raw RGB images of size $64 \times 64 \times 3$. We apply a random-shift augmentation with probability 0.5 and use frame stacking with three consecutive frames. These preprocessing steps are shared across all pixel-based methods and are critical for stable learning in visually complex environments.

**Training and evaluation.** We train PCBF for 1M gradient steps on state-based OGBench tasks and 500K gradient steps on pixel-based OGBench and D4RL tasks. Evaluation is performed every 100K steps using 50 episodes. For OGBench tasks, we report the average success rate over the final three evaluation checkpoints, following the official evaluation protocol. For D4RL Adroit tasks, we report normalized returns at the final training checkpoint.

**Policy extraction.** At deployment, we follow the standard candidate-action protocol used in our FQL-style codebase (Park et al., 2025b): a behavior-cloned proposal policy $\pi_\beta$ samples $K$ candidate actions $\{a_k\}_{k=1}^K$ from $\pi_\beta(\cdot \mid s)$. Each candidate is scored by the mean terminal return under the learned flow,

$$\hat{Q}_\theta(s, a) = \frac{1}{M} \sum_{m=1}^{M} \psi_\theta^1(X_{0,m} \mid s, a), \qquad X_{0,m} \sim \mathcal{N}(0, I), \tag{36}$$

and we execute $\arg\max_{a_k} \hat{Q}_\theta(s, a_k)$. Unless stated otherwise we use $K{=}16$ candidates, matching the "rejection sampling candidates" entry in Table 4.

**Hyperparameters.** The $\lambda$ parameter, which controls the strength of the path-coupled correction, is the primary method-specific hyperparameter in PCBF. We tune $\lambda$ at the domain level using the task marked with $*$ and apply the selected value across all tasks within the same domain. A detailed summary of hyperparameter choices for all methods and domains is provided in Tables 4 and 5.

## E. Computational Cost

PCBF has similar memory and wall-clock cost to Value Flows and is slower than scalar critics; full cost details are in Table 2.

*Table 2.* Training cost on `cube-double-play-task1` (single A100, $10^6$ steps).

| Method | GPU memory (GB) | Wall-clock ($10^3$ seconds) |
|---|---|---|
| IQL | $\sim 12$ | $1.0\times$ |
| FQL | $\sim 14$ | $1.0\times$ |
| Value Flows | $\sim 60$ | $2.4\times$ |
| PCBF (Ours) | $\sim 59.9$ | $2.5\times$ |

## F. Ablation Study

Figure 5 studies the effect of $\lambda$ in PCBF across representative OGBench and D4RL tasks, with all other hyperparameters held fixed. As discussed in Section 4, $\lambda$ interpolates between unbiased sample-based targets ($\lambda = 0$) and variance-reduced corrections ($\lambda > 0$), and performance is highly sensitive to its choice. In several state-based OGBench tasks (e.g., scene-play-task2, cube-double-task2, puzzle-4×4-task4) moderate $\lambda$ values yield the best success rates, whereas for some visual tasks (e.g., visual-antmaze-task1) $\lambda = 0$ performed best. For D4RL Adroit tasks (e.g., hammer-cloned and hammer-expert) different $\lambda$ values can improve normalized returns. This task-dependent variability motivates tuning $\lambda$ at the domain level, as reported in our main experiments.

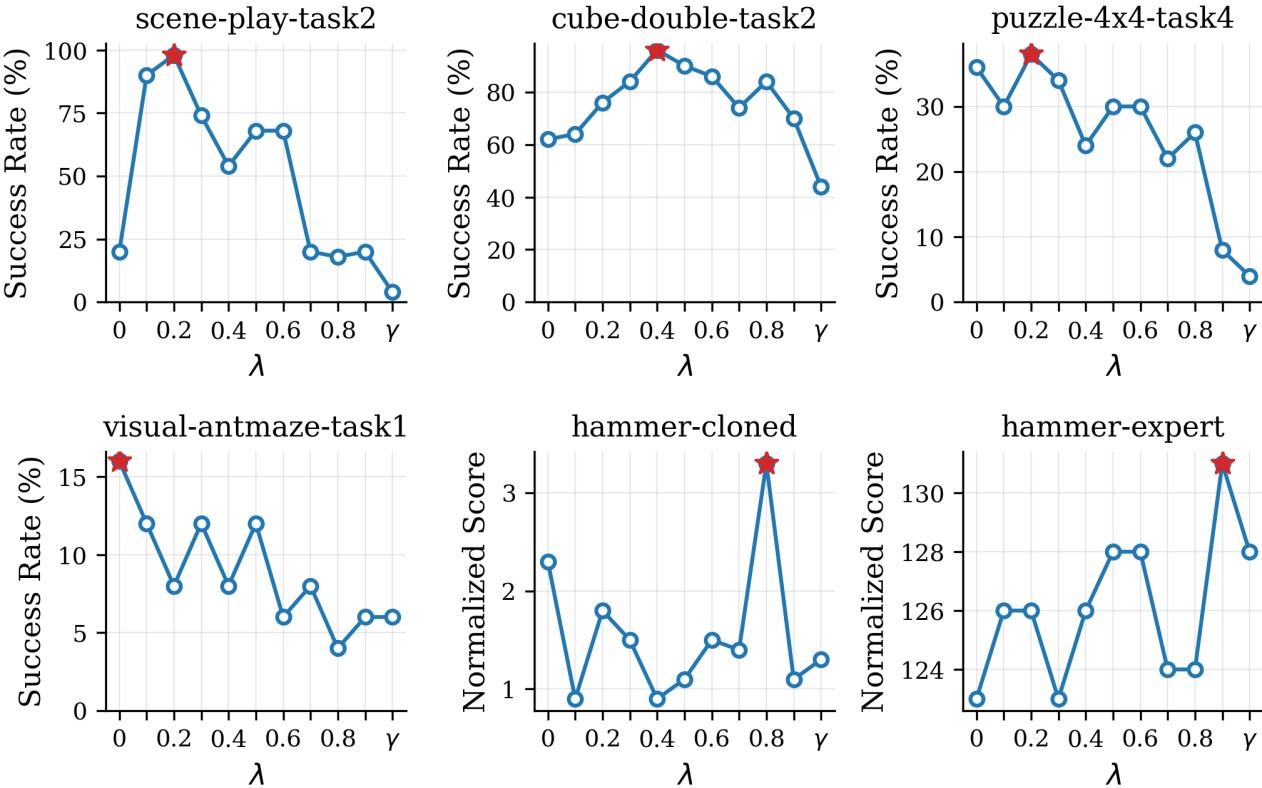

*Figure 5.* **Ablation study of the $\lambda$ parameter in PCBF.** Red stars denote the best-performing $\lambda$ on representative OGBench and D4RL tasks.

# G. Full Experiment Results

*Table 3.* **Full Offline RL Results.** We report performance on the 38 tasks on OGBench and D4RL. The symbol ($*$) indicates the task used for hyperparameter tuning within each domain. Results are averaged over 8 random seeds (4 seeds for pixel-based tasks). Bold numbers denote values that are within $95\%$ of the best performing method on each task.

| | Algorithms | | | | | | |
| --- | --- | --- | --- | --- | --- | --- | --- |
| Task | IQN | CODAC | FloQ | FQL | IQL | Value Flows | PCBF (Ours) |
| cube-double-play-singletask-task1-v0 | $70 \pm 14$ | $80 \pm 11$ | $50 \pm 24$ | $61 \pm 9$ | $27 \pm 5$ | $\mathbf{97 \pm 1}$ | $92 \pm 3$ |
| cube-double-play-singletask-task2-v0 ($*$) | $24 \pm 9$ | $63 \pm 4$ | $72 \pm 15$ | $36 \pm 6$ | $1 \pm 1$ | $\mathbf{76 \pm 7}$ | $\mathbf{74 \pm 7}$ |
| cube-double-play-singletask-task3-v0 | $25 \pm 6$ | $66 \pm 9$ | $57 \pm 14$ | $22 \pm 5$ | $0 \pm 0$ | $73 \pm 4$ | $\mathbf{81 \pm 8}$ |
| cube-double-play-singletask-task4-v0 | $10 \pm 1$ | $13 \pm 2$ | $8 \pm 4$ | $5 \pm 2$ | $0 \pm 0$ | $\mathbf{30 \pm 5}$ | $22 \pm 5$ |
| cube-double-play-singletask-task5-v0 | $81 \pm 8$ | $82 \pm 4$ | $50 \pm 11$ | $19 \pm 10$ | $4 \pm 3$ | $69 \pm 5$ | $\mathbf{84 \pm 3}$ |
| scene-play-singletask-task1-v0 | $\mathbf{100 \pm 0}$ | $99 \pm 0$ | $100 \pm 1$ | $\mathbf{100 \pm 0}$ | $94 \pm 3$ | $99 \pm 0$ | $\mathbf{100 \pm 0}$ |
| scene-play-singletask-task2-v0 ($*$) | $1 \pm 0$ | $85 \pm 4$ | $83 \pm 10$ | $76 \pm 9$ | $12 \pm 3$ | $\mathbf{97 \pm 1}$ | $57 \pm 13$ |
| scene-play-singletask-task3-v0 | $94 \pm 2$ | $90 \pm 3$ | $98 \pm 2$ | $\mathbf{98 \pm 1}$ | $32 \pm 7$ | $94 \pm 2$ | $\mathbf{98 \pm 2}$ |
| scene-play-singletask-task4-v0 | $3 \pm 1$ | $0 \pm 0$ | $9 \pm 7$ | $5 \pm 1$ | $0 \pm 1$ | $7 \pm 17$ | $\mathbf{12 \pm 3}$ |
| scene-play-singletask-task5-v0 | $0 \pm 0$ | $0 \pm 0$ | $0 \pm 0$ | $0 \pm 0$ | $0 \pm 0$ | $0 \pm 0$ | $\mathbf{2 \pm 1}$ |
| puzzle-4x4-play-singletask-task1-v0 | $\mathbf{41 \pm 2}$ | $37 \pm 32$ | $47 \pm 7$ | $34 \pm 8$ | $12 \pm 2$ | $36 \pm 3$ | $38 \pm 6$ |
| puzzle-4x4-play-singletask-task2-v0 | $12 \pm 4$ | $10 \pm 10$ | $21 \pm 6$ | $16 \pm 5$ | $7 \pm 4$ | $\mathbf{27 \pm 5}$ | $23 \pm 5$ |
| puzzle-4x4-play-singletask-task3-v0 | $\mathbf{45 \pm 7}$ | $33 \pm 29$ | $36 \pm 5$ | $18 \pm 5$ | $9 \pm 3$ | $30 \pm 4$ | $40 \pm 4$ |
| puzzle-4x4-play-singletask-task4-v0 ($*$) | $23 \pm 2$ | $12 \pm 10$ | $19 \pm 5$ | $11 \pm 3$ | $5 \pm 2$ | $\mathbf{28 \pm 5}$ | $\mathbf{28 \pm 4}$ |
| puzzle-4x4-play-singletask-task5-v0 | $16 \pm 6$ | $10 \pm 8$ | $16 \pm 7$ | $7 \pm 3$ | $4 \pm 1$ | $13 \pm 2$ | $\mathbf{23 \pm 3}$ |
| cube-triple-play-singletask-task1-v0 ($*$) | $29 \pm 2$ | $9 \pm 5$ | $32 \pm 13$ | $20 \pm 6$ | $4 \pm 4$ | $\mathbf{59 \pm 12}$ | $18 \pm 4$ |
| cube-triple-play-singletask-task2-v0 | $0 \pm 0$ | $\mathbf{1 \pm 0}$ | $0 \pm 0$ | $1 \pm 2$ | $0 \pm 0$ | $0 \pm 0$ | $0 \pm 1$ |
| cube-triple-play-singletask-task3-v0 | $1 \pm 0$ | $0 \pm 0$ | $\mathbf{7 \pm 3}$ | $0 \pm 0$ | $0 \pm 0$ | $\mathbf{7 \pm 3}$ | $1 \pm 1$ |
| cube-triple-play-singletask-task4-v0 | $\mathbf{0 \pm 0}$ | $\mathbf{0 \pm 0}$ | $0 \pm 0$ | $\mathbf{0 \pm 0}$ | $\mathbf{0 \pm 0}$ | $\mathbf{0 \pm 0}$ | $\mathbf{0 \pm 0}$ |
| cube-triple-play-singletask-task5-v0 | $0 \pm 0$ | $0 \pm 0$ | $0 \pm 0$ | $0 \pm 0$ | $1 \pm 1$ | $\mathbf{2 \pm 1}$ | $1 \pm 1$ |
| pen-cloned-v1 | $80 \pm 11$ | $76 \pm 2$ | $72 \pm 5$ | $74 \pm 11$ | $\mathbf{83}$ | $73 \pm 5$ | $78 \pm 5$ |
| pen-expert-v1 | $118 \pm 19$ | $136 \pm 2$ | $140 \pm 8$ | $\mathbf{142 \pm 6}$ | $128$ | $117 \pm 3$ | $131 \pm 5$ |
| door-cloned-v1 | $0 \pm 0$ | $0 \pm 0$ | $\mathbf{3 \pm 2}$ | $2 \pm 1$ | $\mathbf{3}$ | $0 \pm 0$ | $1 \pm 0$ |
| door-expert-v1 | $105 \pm 0$ | $104 \pm 0$ | $104 \pm 0$ | $104 \pm 1$ | $\mathbf{107}$ | $104 \pm 1$ | $\mathbf{106 \pm 0}$ |
| hammer-cloned-v1 | $0 \pm 0$ | $6 \pm 0$ | $10 \pm 8$ | $\mathbf{11 \pm 9}$ | $2$ | $1 \pm 0$ | $2 \pm 1$ |
| hammer-expert-v1 | $121 \pm 7$ | $126 \pm 1$ | $125 \pm 2$ | $125 \pm 3$ | $\mathbf{129}$ | $125 \pm 5$ | $\mathbf{126 \pm 2}$ |
| relocate-cloned-v1 | $0 \pm 0$ | $0 \pm 0$ | $0 \pm 0$ | $0 \pm 0$ | $\mathbf{2}$ | $0 \pm 0$ | $0 \pm 0$ |
| relocate-expert-v1 | $103 \pm 0$ | $103 \pm 2$ | $108 \pm 2$ | $107 \pm 1$ | $106$ | $102 \pm 2$ | $\mathbf{109 \pm 1}$ |
| visual-antmaze-teleport-navigate-singletask-task1-v0 ($*$) | $2 \pm 1$ | – | – | $2 \pm 1$ | $5 \pm 2$ | $\mathbf{10 \pm 4}$ | $8 \pm 2$ |
| visual-antmaze-teleport-navigate-singletask-task2-v0 | $7 \pm 3$ | – | – | $6 \pm 1$ | $10 \pm 2$ | $17 \pm 5$ | $\mathbf{19 \pm 4}$ |
| visual-antmaze-teleport-navigate-singletask-task3-v0 | $6 \pm 4$ | – | – | $9 \pm 4$ | $7 \pm 7$ | $16 \pm 3$ | $\mathbf{18 \pm 4}$ |
| visual-antmaze-teleport-navigate-singletask-task4-v0 | $4 \pm 2$ | – | – | $9 \pm 1$ | $4 \pm 6$ | $16 \pm 5$ | $\mathbf{18 \pm 5}$ |
| visual-antmaze-teleport-navigate-singletask-task5-v0 | $2 \pm 1$ | – | – | $1 \pm 1$ | $2 \pm 1$ | $\mathbf{8 \pm 2}$ | $8 \pm 3$ |
| visual-cube-double-play-singletask-task1-v0 ($*$) | $4 \pm 1$ | – | – | $23 \pm 4$ | $34 \pm 23$ | $\mathbf{35 \pm 2}$ | $10 \pm 3$ |
| visual-cube-double-play-singletask-task2-v0 | $0 \pm 0$ | – | – | $0 \pm 0$ | $3 \pm 1$ | $\mathbf{4 \pm 2}$ | $0 \pm 0$ |
| visual-cube-double-play-singletask-task3-v0 | $0 \pm 0$ | – | – | $0 \pm 0$ | $7 \pm 4$ | $\mathbf{11 \pm 2}$ | $0 \pm 0$ |
| visual-cube-double-play-singletask-task4-v0 | $0 \pm 0$ | – | – | $\mathbf{4 \pm 2}$ | $2 \pm 1$ | $2 \pm 1$ | $0 \pm 0$ |
| visual-cube-double-play-singletask-task5-v0 | $1 \pm 1$ | – | – | $4 \pm 1$ | $11 \pm 2$ | $\mathbf{13 \pm 3}$ | $3 \pm 1$ |

# H. Experiment Details

We implement PCBF and all baselines using JAX (Bradbury et al., 2018), with implementations adapted from the FQL codebase (Park et al., 2025b).

## H.1. Environments and Datasets

We evaluate PCBF on offline reinforcement learning benchmarks from OGBench (Park et al., 2025a) and D4RL (Fu et al., 2020), which provide diverse tasks with long-horizon dependencies, semi-sparse rewards, and multimodal return distributions.

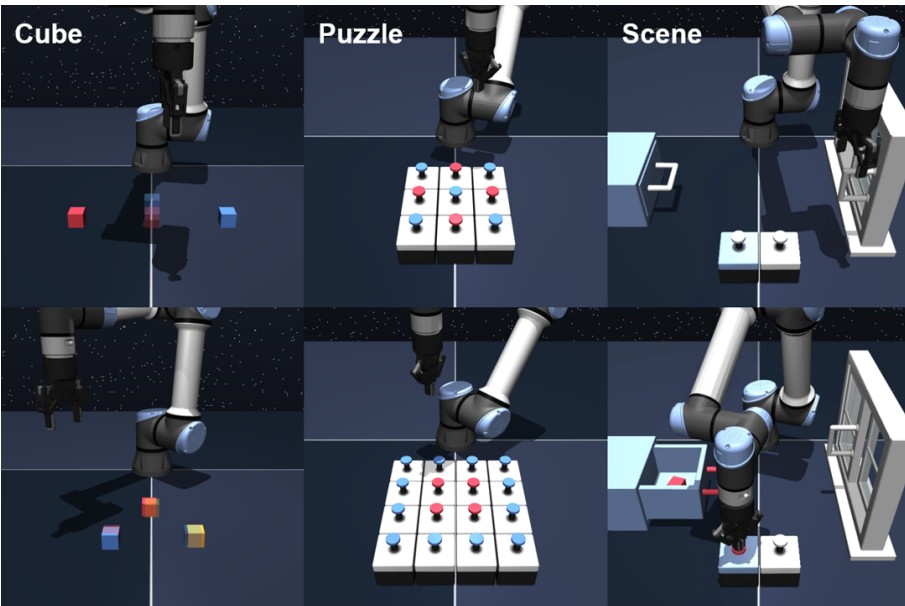

*Figure 6.* OGBench Tasks

**OGBench ([Park et al., 2025a](#)).** OGBench is originally designed for offline goal-conditioned reinforcement learning. Following prior work, we adopt its single-task variants ("-singletask") to benchmark standard reward-maximizing offline RL methods. In each environment, five predefined evaluation goals are provided, yielding five corresponding single-task variants (from -singletask-task1 to -singletask-task5). Given a fixed evaluation goal, transitions in the dataset are labeled with a semi-sparse reward function that reflects task progress. For state-based and pixel-based OGBench, we use the following environments:

- **State-based datasets**
    - cube-double-play-v0
    - cube-triple-play-v0
    - scene-play-v0
    - puzzle-4x4-play-v0

- **Pixel-based datasets**
    - visual-cube-double-play-v0
    - visual-antmaze-teleport-navigate-v0

We choose these environments to cover a diverse range of challenges in long-horizon robotic manipulation and compositional reasoning, following the manipulation suite provided by OGBench ([Park et al., 2025a](#)). The cube, scene, and puzzle environments are robot-arm manipulation tasks built on a 6-DoF robotic arm, where agents must interact with multiple objects over extended horizons. The cube environments involve multi-stage pick-and-place manipulation of colored blocks and require arranging objects into target configurations through sequences of coordinated actions. The scene-play environment further increases complexity by involving multiple interactive objects—such as drawers, locks, and movable blocks—where solving a task often requires executing a specific sequence of dependent subtasks, with the longest tasks involving up to eight atomic operations. The puzzle-4×4 environment corresponds to robotic variants of the Lights Out puzzle and is designed to test combinatorial generalization, as agents must reason over button configurations and long sequences of interactions to reach the desired goal state. All state-based manipulation tasks follow the standard play-style dataset setting in OGBench, where datasets are collected from scripted policies performing random interactions rather than goal-directed demonstrations. As a result, the datasets exhibit high suboptimality and require agents to effectively stitch partial behaviors into coherent long-horizon solutions. We evaluate these environments using their single-task variants,

where each environment provides five predefined evaluation goals and rewards are semi-sparse, reflecting incremental progress toward task completion. In addition to state-based environments, we include pixel-based OGBench tasks that require learning directly from raw RGB observations of size $64 \times 64 \times 3$. These visual variants introduce partial observability and additional perceptual complexity, further challenging return modeling and policy learning. Due to the significantly higher computational cost of pixel-based training, we evaluate a representative subset of visual tasks in our experiments. For all OGBench tasks, we use the standard dataset types (play for manipulation and navigate for navigation) and report binary task success rates (in percentage), following the original OGBench evaluation protocol.

**D4RL** (Fu et al., 2020). We further evaluate PCBF on tasks from the **D4RL** benchmark, which is widely used for studying offline reinforcement learning. In our experiments, the Adroit manipulation tasks require dexterous control with a high-dimensional 24-DoF action space. These tasks involve contact-rich interactions and long-horizon dependencies, providing a complementary evaluation setting to OGBench manipulation environments. Following the standard D4RL evaluation protocol, performance on Adroit tasks is reported using normalized returns. We use the following 8 adroit tasks:

- **D4RL Adroit tasks**

  - pen-cloned-v1
  - pen-expert-v1
  - door-cloned-v1
  - door-expert-v1
  - hammer-cloned-v1
  - hammer-expert-v1
  - relocate-cloned-v1
  - relocate-expert-v1

### H.2. Methods and Hyperparameters

We compare PCBF against a set of strong offline reinforcement learning baselines, covering both scalar value-based and distributional methods. All methods use the same network architectures and discount factors.

- **IQL** (Kostrikov et al., 2022): Implicit Q-Learning is a scalar value-based offline RL method that learns conservative Q-functions via expectile regression and extracts actions within the support of the offline dataset.

- **FQL** (Park et al., 2025b): Flow Q-Learning is a scalar value-based method that employs a flow-based policy to maximize Q-values learned through standard temporal-difference updates, together with behavioral regularization.

- **FloQ** (Agrawalla et al., 2026): Flow-matching Q-functions use a state-action-conditioned velocity field to parameterize scalar Q-values through numerical integration. FloQ trains this velocity field with TD-bootstrapped flow-matching targets, enabling iterative compute scaling for value estimation rather than modeling the full return distribution.

- **IQN** (Dabney et al., 2018a): Implicit Quantile Networks approximate the return distribution by predicting quantile values at randomly sampled quantile fractions using quantile regression.

- **CODAC** (Ma et al., 2021): CODAC extends distributional RL with conservative regularization to mitigate overestimation, combining quantile-based critics with offline constraints.

- **Value Flows** (Dong et al., 2026): Value Flows is a flow-based distributional RL method that models the full return distribution using flow matching and learns value functions via continuous density transformations. Value Flows employs a weighted sampling procedure for time $t$ during training, whereas PCBF uses uniform time sampling.

- **PCBF (ours)**: Path-Coupled Bellman Flows model return distributions with flow matching using source-consistent Bellman-coupled paths, shared-noise coupling, and the $\lambda$-target of Section 4. Details are in Appendix D; hyperparameters are in Tables 4–5.

*Table 4.* **Common hyperparameters for PCBF and baselines.**

| Hyperparameter | Value |
|---|---|
| optimizer | Adam (Kingma & Ba, 2015) |
| batch size | 256 |
| learning rate | $3 \times 10^{-4}$ |
| MLP hidden layer sizes | (512, 512, 512, 512) |
| MLP activation function | GELU (Hendrycks & Gimpel, 2016) |
| use actor layer normalization | Yes |
| use value layer normalization | Yes |
| number of flow steps in the Euler method | 10 |
| number of candidates in rejection sampling | 16 |
| target network update coefficient | $5 \times 10^{-3}$ |
| number of Q ensembles | 2 |
| image encoder | small IMPALA encoder (Espeholt et al., 2018; Park et al., 2025b) |
| image augmentation method | random cropping |
| image frame stack | 3 |

*Table 5.* **Hyperparameters for PCBF and baselines.** Hyperparameters are tuned per domain on the task marked with $*$ in the OGBench benchmarks. Entries marked with $-$ indicate not applicable or inherited settings (Park et al., 2025b). The discount factor $\gamma$ is shared across methods within each domain.

| Domain or task | Discount $\gamma$ | IQL $\alpha$ | FQL $\alpha$ | IQN $\kappa$ | IQN $\alpha$ | CODAC $\alpha$ | CODAC $\lambda$ | PCBF $\lambda$ |
|---|---|---|---|---|---|---|---|---|
| cube-double-play | 0.995 | 0.3 | 100 | 0.95 | 300 | 1 | 3 | 0.4 |
| cube-triple-play | 0.995 | 10 | 100 | 0.95 | 1000 | 3 | 0.03 | 0.995 |
| puzzle-4x4-play | 0.99 | 3 | 300 | 0.95 | 1000 | 3 | 100 | 0.2 |
| scene-play | 0.99 | 10 | 100 | 0.95 | 100 | 1 | 0.3 | 0.2 |
| pen-cloned | 0.99 | $-$ | 10000 | 0.8 | 10000 | 3 | 1 | 0.7 |
| pen-expert | 0.99 | $-$ | 3000 | 0.8 | 10000 | 3 | 0.01 | 0.7 |
| door-cloned | 0.99 | $-$ | 30000 | 0.9 | 30000 | 10 | 0.3 | 0.5 |
| door-expert | 0.99 | $-$ | 30000 | 0.9 | 10000 | 10 | 0.3 | 0.99 |
| hammer-cloned | 0.99 | $-$ | 10000 | 0.8 | 10000 | 3 | 0.3 | 0.8 |
| hammer-expert | 0.99 | $-$ | 30000 | 0.8 | 10000 | 10 | 1 | 0.9 |
| relocate-cloned | 0.99 | $-$ | 30000 | 0.9 | 30000 | 3 | 0.01 | 0.99 |
| relocate-expert | 0.99 | $-$ | 30000 | 0.9 | 10000 | 3 | 0.1 | 0.99 |
| visual-antmaze-teleport-navigate | 0.99 | 1 | 100 | $-$ | $-$ | 0.3 | 0.03 | 0 |
| visual-cube-double-play | 0.995 | 0.3 | 100 | $-$ | $-$ | 1 | 0.3 | 0.9 |

# I. Additional Results and Analysis

## I.1. Toy Environment Details

To directly evaluate whether PCBF learns accurate return distributions, we study a set of analytically tractable toy environments with known return structure. In contrast to large-scale control benchmarks such as OGBench, which necessarily involve action selection and policy optimization, these toy environments admit return laws that can be derived exactly or characterized in closed form. This allows us to access distributional accuracy unambiguously, rather than inferring it indirectly from control performance.

The environments are chosen to expose different forms of stochasticity in the Bellman recursion while remaining simple enough to permit precise interpretation. We consider:

- (i) **Solitaire Dice** environment, which produces long-tailed and high-variance returns through stochastic termination while remaining free of action-dependent dynamics.

- (ii) **Single-state Bernoulli MRP** with a closed-form return distribution, which serves as a sanity check because its

discounted return is exactly uniformly distributed over a bounded interval, enabling direct comparison between learned return distributions and the analytic ground truth.

- (iii) **Discrete Monte Carlo Markov chain** with finite-horizon returns induced by nearest-neighbor dynamics, which enables controlled analysis of Bellman target variability as a function of effective horizon length.

Together, these environments provide a principled testbed for analyzing how the $\lambda$-parameterized control variate in PCBF influences training behavior and return-distribution accuracy as horizon length and return variance increase, and for directly comparing the fidelity of learned return distributions across PCBF and baseline distributional RL methods against known ground-truth laws.

**Solitaire Dice.** We first consider a stochastic termination process adapted from (Bellemare et al., 2023), which we refer to as the *Solitaire Dice* environment. The environment consists of repeatedly rolling a single fair six-sided die. If a 1 is rolled, the episode terminates immediately; otherwise, the agent receives a reward of 1 and the game continues. The environment contains no actions, and the transition dynamics are entirely driven by the die outcome.

Let $T$ denote the number of non-terminating rolls before the first terminating outcome 1. The undiscounted return is given by

$$G = \sum_{t=0}^{\infty} R_t = \underbrace{1 + 1 + \cdots + 1}_{T \text{ times}},$$

which is an integer-valued random variable taking values in $\mathbb{N} = \{0, 1, 2, \dots\}$. Since each roll independently terminates the episode with probability $\frac{1}{6}$, the return follows a geometric distribution,

$$\mathbb{P}(G = k) = \tfrac{1}{6} \left(\tfrac{5}{6}\right)^k, \qquad k \in \mathbb{N},$$

corresponding to observing $k$ non-terminating outcomes before the first terminating roll.

When a discount factor $\gamma \in (0, 1)$ is introduced, the return becomes

$$G_\gamma = \sum_{t=0}^{\infty} \gamma^t R_t,$$

Conditioned on $T = k$, this sum evaluates to the partial geometric series

$$G_\gamma = \sum_{t=0}^{k-1} \gamma^t = \frac{1 - \gamma^k}{1 - \gamma},$$

Importantly, discounting changes the *support* of the return distribution but not the associated probabilities: for all $k \geq 0$,

$$\mathbb{P}\left(G_\gamma = \frac{1 - \gamma^k}{1 - \gamma}\right) = \tfrac{1}{6} \left(\tfrac{5}{6}\right)^k.$$

This environment induces a highly skewed, long-tailed return distribution driven entirely by stochastic termination. As such, it provides a controlled setting for studying Bellman target variance arising from random episode lengths and for evaluating the stability of distributional learning methods under heavy-tailed returns.

**Bernoulli.** We next consider a single-state, single-action Markov decision process with purely stochastic rewards, adapted from the classical example in (Bellemare et al., 2023). The state space is $\mathcal{X} = \{x\}$ and the action space is $\mathcal{A} = \{a\}$. The initial distribution is $\xi_0 = \delta_x$, and the transition kernel is deterministic,

$$P(x \mid x, a) = 1,$$

so the system remains in the same state at all times. The reward process $\{R_t\}_{t \geq 0}$ is i.i.d. with

$$\mathbb{P}(R_t = 1 \mid x, a) = \mathbb{P}(R_t = 0 \mid x, a) = \tfrac{1}{2},$$

We fix the discount factor to $\gamma = \frac{1}{2}$. The discounted return is therefore

$$G \;=\; \sum_{t=0}^{\infty} \gamma^t R_t \;=\; R_0 + \tfrac{1}{2}R_1 + \tfrac{1}{4}R_2 + \cdots ,$$

A key observation is that $G$ admits a binary expansion

$$G \;=\; R_0.R_1 R_2 \ldots \quad ,$$

where the expression denotes the binary expansion of a number in $[0,2]$ and each digit is an independent Bernoulli random variable. As a consequence, the support of $G$ is the interval $[0,2]$, with $0$ corresponding to the infinite sequence of zeros and $2$ corresponding to the infinite sequence of ones. Moreover, for any dyadic interval $[a,b] \subset [0,2]$ whose endpoints admit finite binary expansions, the probability mass satisfies

$$\mathbb{P}(G \in [a,b]) = \tfrac{b-a}{2},$$

This property uniquely characterizes the uniform distribution on $[0,2]$, implying that the return distribution is exactly

$$G \sim \mathrm{Unif}[0,2].$$

This environment provides a rare example of an MDP with a closed-form return distribution despite infinite-horizon bootstrapping. Because the dynamics are trivial and all stochasticity arises solely from the reward sequence, it serves as a clean sanity check for distributional Bellman consistency and for analyzing variance-reduction effects in PCBF without confounding effects from state transitions, exploration, or function approximation.

**Discrete Monte Carlo Chain.** We next consider a finite-state Markov reward process adapted from the discrete nearest-neighbor Markov chains studied in Cheng & Weare (2024). The environment consists of a one-dimensional Markov chain on a discrete state space $\mathcal{X} = \{0, 1, \ldots, n-1\}$ with no actions. States $0$ and $n-1$ are absorbing terminal states, and episodes are initialized from a non-terminal state in $\{1, \ldots, n-2\}$.

The transition dynamics follow a nearest-neighbor structure with state-dependent probabilities. Let $p : \mathcal{X} \to \mathbb{R}_{>0}$ be defined by

$$p(i) \;\propto\; \exp\left(\frac{n-1}{4\pi} \cos\left(\frac{4\pi(i-1)}{n-1}\right)\right),$$

and define the transition kernel for $i \in \{1, \ldots, n-2\}$ by

$$P(i, i \pm 1) \;\propto\; \frac{p(i \pm 1)}{p(i) + p(i \pm 1)}, \qquad P(i, i) = 1 - P(i, i-1) - P(i, i+1),$$

with $P(0,0) = P(n-1, n-1) = 1$ (absorbing boundaries). This construction induces a multi-well potential landscape in which local transitions remain stable while global escape times grow with $n$.

The reward function is deterministic: the agent receives a reward of $1$ at each non-terminal transition and $0$ upon entering a terminal state. The resulting return is

$$G \;=\; \sum_{t=0}^{T-1} 1.$$

where $T$ is the (finite) first hitting time of the terminal set. By construction, the return distribution is fully determined by the transition dynamics and the episode horizon.

This environment provides a controlled finite-horizon testbed in which Bellman targets accumulate stochasticity through repeated transitions rather than unbounded reward support. It enables direct examination of how Bellman target variance scales with effective horizon length, and facilitates precise comparison of learned return distributions across PCBF and baseline methods in a setting where the underlying return law is exactly defined.

## I.2. Internal Analysis: Variance Reduction and Stability

**Variance reduction during training.** Figure 7 reports the within-run standard deviation of the Bellman velocity regression loss. As $\lambda$ increases, this variability consistently decreases. This demonstrates that the $\lambda$-parameterized control variate effectively dampens the noise in gradient updates caused by stochastic bootstrapping, validating its role as a variance-reduction mechanism.

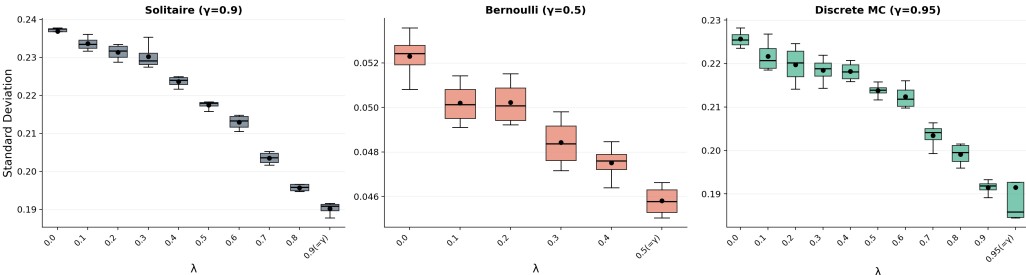

*Figure 7.* **Variance reduction via $\lambda$-parameterized control variates.** Larger $\lambda$ yields smoother loss trajectories (lower standard deviation), demonstrating effective variance reduction in Bellman targets.

**Bias–variance trade-off.** While increasing $\lambda$ reduces optimization variance, Figure 8 illustrates the resulting trade-off in distributional accuracy. In the Bernoulli environment ($\gamma = 0.5$), the Wasserstein distance remains low for $\lambda \leq 0.3$ but increases as $\lambda \to \gamma$. In the more challenging Discrete Monte Carlo environment ($\gamma = 0.95$), we observe a "sweet spot" at moderate $\lambda$, where the benefit of variance reduction outweighs the bias, yielding the lowest approximation error.

## I.3. Comparative Analysis: PCBF vs. Value Flows

We compare PCBF against Value Flows (Dong et al., 2026) to highlight the impact of our proposed boundary conditions versus the Bellman consistency used in Value Flows (controlled by the *dcfm* coefficient).

**Sensitivity to Consistency Regularization.** Figure 8 provides a direct comparison of hyperparameter sensitivity between our method (PCBF) and Value Flows. The results reveal a sharp contrast in stability:

- **Value Flows Instability (Orange):** Increasing the DCFM coefficient (`dcfm`) systematically degrades performance across all tasks. This sensitivity is most extreme in the *Discrete MC* environment (Right), where Wasserstein distance starts high ($\approx 1.5$) and explodes to $> 4.0$ as `dcfm` $\to 1$. This is consistent with overweighting a full-$t$ DCFM-style self-consistency term relative to the BCFM anchoring that stabilizes the practical Value Flows objective.

- **PCBF Robustness (Blue):** In contrast, our Control Variate formulation ($\lambda$) demonstrates remarkable stability. In *Discrete MC*, PCBF maintains a consistently low Wasserstein distance ($\approx 0.5$) across the entire range of $\lambda \in [0, 0.95]$, effectively ignoring the bias that plagues Value Flows. Similarly, in *Solitaire* and *Bernoulli*, PCBF outperforms Value Flows at nearly all coefficient magnitudes, showing that our method reduces variance without introducing significant bias into the terminal distribution.

**Detailed Distributional Comparison.** Figure 9 presents a granular comparison of Cumulative Distribution Functions (CDFs) and Wasserstein distances. Across all six evaluation settings, PCBF consistently achieves the lowest or near-lowest Wasserstein distance to the ground truth.

- **Long Horizons:** In the Discrete MC environment ($S = 5$), VF with dcfm$= 1$ degrades to a Wasserstein distance of $6.856$ due to bias accumulation, while PCBF maintains $0.586$.

- **Tail Accuracy:** The CDF plots reveal that VF systematically underestimates return variance (producing overly concentrated distributions), whereas PCBF closely tracks the reference tails.

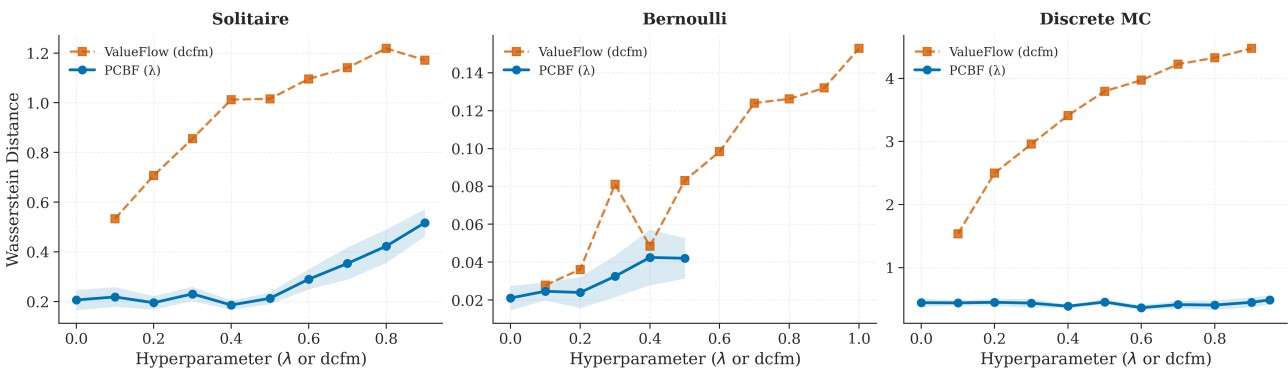

*Figure 8.* **Hyperparameter Sensitivity Analysis (PCBF vs. Value Flows)** We compare the impact of increasing the regularization coefficient on distributional accuracy (Wasserstein Distance). **Orange (Dashed):** Increasing the Value Flows consistency coefficient (dcfm) causes rapid performance degradation, particularly in complex environments like Discrete MC. **Blue (Solid):** Our PCBF Control Variate ($\lambda$) remains robust, maintaining low Wasserstein distances and high stability across a wide range of hyperparameter values, effectively decoupling variance reduction from distributional bias.

### I.4. Visual Analysis of Transport Dynamics

Furthermore, we visually analyze the learned transport maps to verify that PCBF captures the correct distributional geometry. Figure 10 presents the learned flows for the Discrete Monte Carlo environment across states $s = 1, \ldots, 20$. The bottom panels illustrate the trajectories that deterministically transport the base noise to the complex, state-dependent target return distributions. The resulting flow-transported densities (top panels, blue) align tightly with ground-truth Monte Carlo histograms (black dashed) across the entire state space. The visualization confirms that the method robustly approximates the true return law $Z^\pi(x, a)$ across diverse structures of stochasticity, justifying the low Wasserstein errors observed quantitatively.

### I.5. Pathwise Bellman Residual and Discretization

PCBF enforces the Bellman endpoint at $t{=}1$ by construction, but training uses a finite-step Euler solver. Let $\hat{Z}_t^s$ and $\hat{Z}_t^{s'}$ denote numerically integrated current and successor paths with $N$ function evaluations (NFE), and let $\tilde{\gamma} = \gamma(1 - d)$ on the transition. We report the corrected residual

$$r_{\text{corr}}(t, N) := \mathbb{E}\left[\left|\hat{Z}_t^s - \left(tR + \tilde{\gamma}\,\hat{Z}_t^{s'} + (1 - t)(1 - \tilde{\gamma})X_0\right)\right|\right], \tag{37}$$

which compares the integrated path to the *closed-form* PCBF interpolation using the same integrated successor path (nonterminal transitions, $d{=}0$). We sweep $t \in [0, 1]$ and $N \in \{4, 8, 16, 32\}$ on Solitaire Dice and compare shared-noise PCBF to an *independent-noise* ablation where the successor path uses an independent $X_0'$ while holding the solver budget fixed. Figure 11 shows that shared-noise coupling yields smaller $r_{\text{corr}}$ across $(t, N)$, indicating that coupling reduces mismatch introduced by coarse discretization.

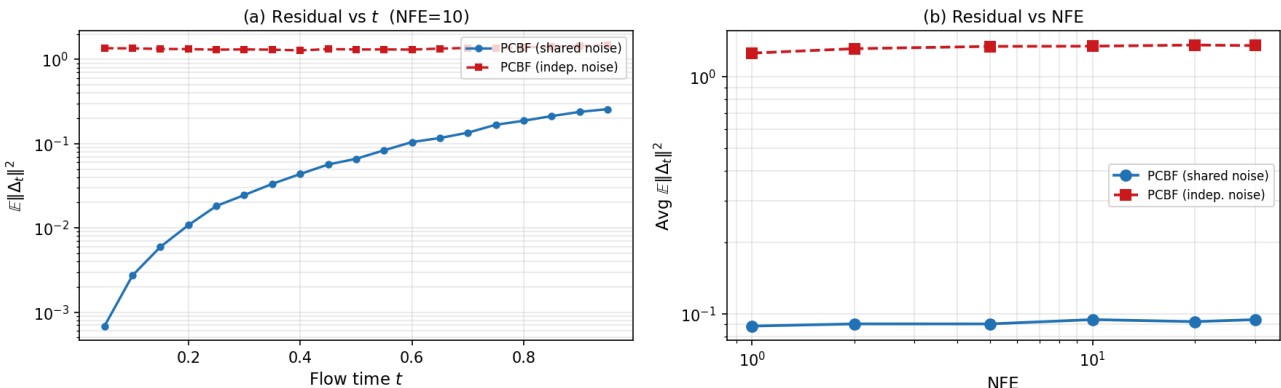

*Figure 11.* Corrected Bellman residual $r_{\mathrm{corr}}(t, N)$ on Solitaire Dice. Shared-noise PCBF (blue) maintains lower residuals than independent-noise coupling (orange) across times and budgets.

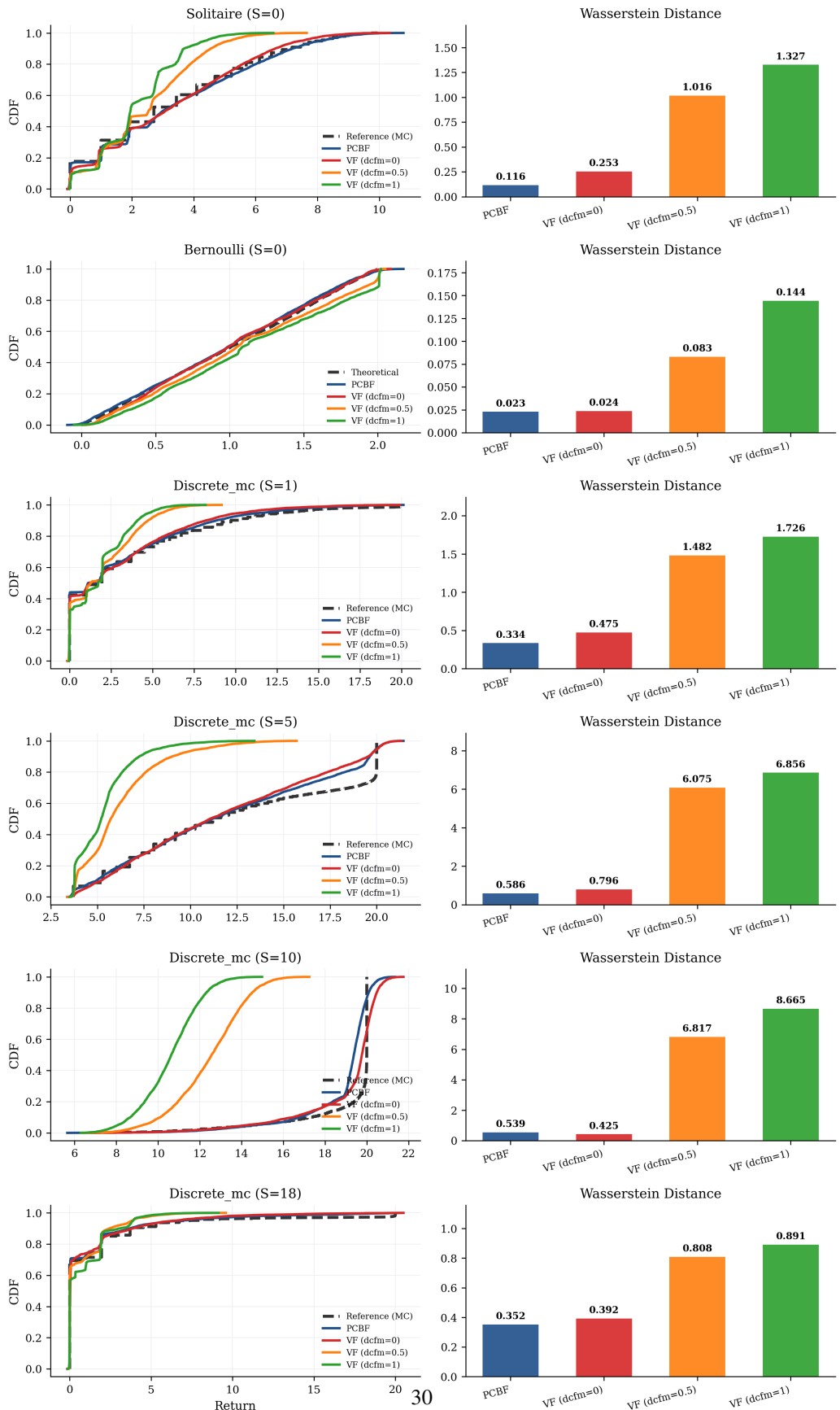

*Figure 9.* **Full Distributional accuracy comparison.** PCBF (blue) consistently tracks the ground-truth CDF (dashed black) more accurately than Value Flows (red/green), particularly in high-variance regimes.

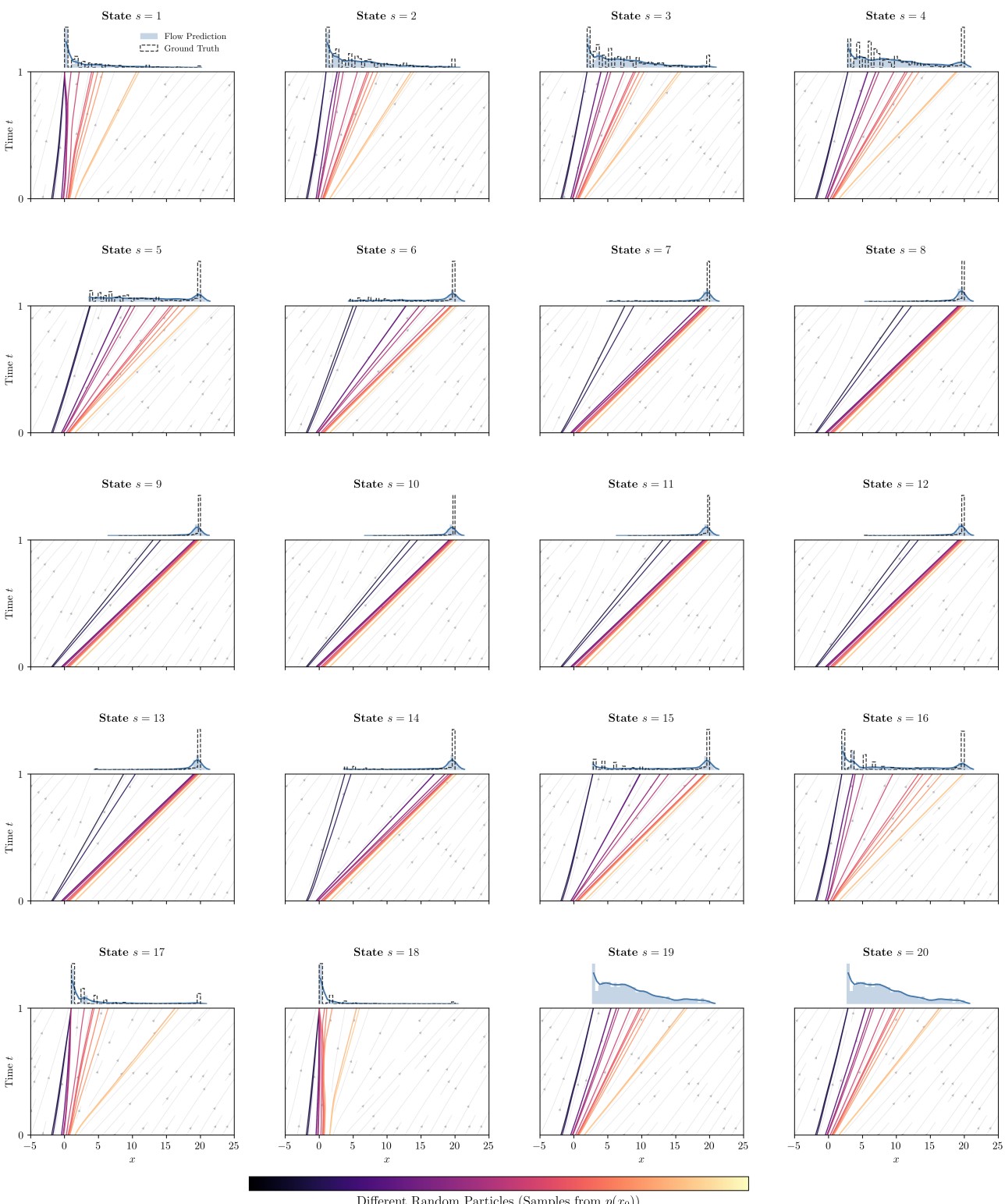

*Figure 10.* **Distributional Flow Analysis on the Discrete MC Environment.** We visualize the learned PCBF return distributions across states $s = 1$ to $s = 20$. The estimated probability density of the flow-transported samples (blue filled) is compared against Ground Truth Monte Carlo rollouts(black dashed lines). Characteristic flow trajectories transporting random noise samples ($t = 0$) to the target return distribution ($t = 1$) over flow time. Trajectory colors distinguish individual particles sampled from the base distribution $p(x_0)$, illustrating how the model maps stochastic noise to specific return outcomes.

