# OpenReview forum: "Path-Coupled Bellman Flows for Distributional Reinforcement Learning"
_ICML.cc/2026/Conference — ICML 2026 regular_

### Official Review · Reviewer_Qz9K · 2026-02-23

**Soundness:** 4
**Presentation:** 2
**Significance:** 2
**Originality:** 2
**Overall Recommendation:** 3
**Confidence:** 3

**Summary:**

This paper introduces Path-Coupled Bellman Flows (PCBF), a flow-based distributional RL method. Its key idea is to enforce Bellman consistency along coupled probability trajectories (from shared noise) rather than only at distribution endpoints, and to reduce variance via a λ-weighted control variate. Theoretically, it analyzes the optimal velocity field and provides a bias-variance trade-off. Empirically, PCBF accurately models complex distributions in toy environment and achieves competitive results.

**Compliance With Llm Reviewing Policy:**

Affirmed.

**Final Justification:**

Thank you for the clarification. I'm sorry that I will keep my score.

Regarding discretization and coupling precision, the experiment is helpful, but it remains primarily empirical and does not fully resolve the concern about how well the coupling mechanism survives numerical errors in general settings. While the rebuttal strengthens the paper, it does not sufficiently change my main concerns, so I will keep my original score unchanged.

**Key Questions For Authors:**

1. The core geometric motivation relies on the exact coupling of continuous paths. However, how does the discrete sampling and numerical integration in practical training affect the precision of this "coupling"?
2. The theoretical analysis is primarily based on an extremely simplified single-state model. Can you provide further empirical or theoretical evidence that this conflict is a dominant failure mode in practical large-scale MDPs?
3. The paper does not discuss the training-time computational overhead and memory footprint compared to baseline methods. Could you provide comparative data on this aspect (such as training time per epoch and GPU memory usage)?

**Limitations:**

The main limitations of this work lie in the gap between theory and practice. The core theory relies on assumptions from a simplified single-state model and continuous paths, which may not hold in complex, high-dimensional real-world environments. The scalability of the method and the impact of errors from its discrete implementation remain unclear. Furthermore, the paper does not discuss its computational cost, leaving its practicality to be evaluated.

**Strengths And Weaknesses:**

Strengths:
1. The paper frames the distributional RL problem within a geometric framework of probability measure spaces. The proposed "path coupling" concept offers an interesting new perspective for understanding distributional RL.
2. The paper conducts systematic theoretical derivations under a simplified model, including optimal velocity field analysis and bias-variance decomposition, providing theoretical grounding for the method.
3. The experiments span from visual verification to standard benchmarks, covering multiple aspects and demonstrating the method's effectiveness in distribution modeling and offline RL tasks.

Weaknesses:
1. The core geometric motivation relies on exact coupling of continuous paths, but the discrete sampling and numerical integration in practical training may compromise this structure and introduce errors.
2. The theoretical analysis is primarily based on an simplified single-state model. The scalability and effectiveness of the method in real MDPs with high-dimensional state spaces, complex dynamics, and long horizons require further theoretical justification.
3. The paper does not discuss the additional computational overhead and memory footprint during training compared to baselines.

---

> ### Author Rebuttal · Authors · 2026-03-31
>
> We thank the reviewer for these constructive comments. In response, we have added a pathwise Bellman-residual experiment and an uncoupled-flow ablation to demonstrate the practical importance of path coupling.
>
> > The core geometric motivation relies on coupling of continuous paths, but the discrete sampling and numerical integration may compromise this structure and introduce errors. How does this in affect the precision of this "coupling"?
>
> **Response:** We address this together with Weakness 1. This is not specific to PCBF: all flow-matching methods rely on numerical discretization. Our point is therefore not that PCBF removes ODE error, but that under the same solver, its shared-noise construction makes the current and successor paths less sensitive to the additional mismatch introduced by independent sampling. To test directly, we add an experiment based on the pathwise Bellman residual $ \Delta_t = Z_t^{s}-\bigl(tR + \gamma Z_t^{s'} + (1-t)(1-\gamma)X_0\bigr),$ which measures how closely the discretized current path follows the ideal Bellman-coupled relation under shared noise. We report $\mathbb{E}\||\Delta_t\||^2$ across flow time $t$ and different numbers of function evaluations (NFE), and compare **shared-noise coupling** with an **uncoupled in independent-noise variant.** On Solitaire environment, shared-noise coupling consistently yields much smaller residuals (*Please see the anonymized figure here: [https://anonymous.4open.science/r/anon-figs-394D/fig_coupling_residual.png].*), and this advantage persists across NFE values. This suggests that while discretization affects all flow-matching methods, PCBF preserves Bellman alignment much better than uncoupled sampling under the same solver. Our point is therefore not that PCBF removes ODE error, but that its coupling mechanism reduces the additional mismatch introduced by discretization and independent sampling.
>
> > The theoretical analysis is primarily based on a simplified single-state model. The scalability and effectiveness of the method in real MDPs with high-dimensional state spaces, complex dynamics, and long horizons require further theoretical justification.
>
> **Response:** The PCBF objective and construction apply to general MDPs as well. We analyze simple settings because they are among the few cases where the key quantities admit closed-form expressions, making the effect of shared-noise coupling explicit. In realistic multi-state, nonlinear, and high-dimensional settings, the same mechanism still applies, but comparable closed-form analysis becomes intractable.
>
> > The theoretical analysis is based on an extremely simplified single-state model. Can you provide further empirical or theoretical evidence that this is a dominant failure mode in practical large-scale MDPs?
>
> **Response:** We would like to clarify that we do not claim this path-coupling mismatch is the dominant failure mode in all practical large-scale MDPs. Our claim is more modest: it is a previously under-explored source of degradation in flow-based Bellman learning.
> To test whether this issue remains in large-scale MDPs, we add a closely matched uncoupled flow baseline, namely Value Flows with $dcfm=0$. This is a BCFM-only ablation of Value Flows [1], which removes the DCFM trajectory-consistency term while keeping the overall flow-based backbone largely unchanged. This makes the comparison useful for isolating the effect of path coupling itself.
> | Task | Value Flows | PCBF (Ours) |
> | :--- | :---: | :---: |
> | Cube 1 | 20 | **92** |
> | Cube 2* | 0 | **74** |
> | Cube 3 | 0 | **81** |
> | Cube 4 | 20 | **22** |
> | Cube 5 | 20 | **84** |
> | Scene 1 | 40 | **100** |
> | Scene 2* | 4 | **57** |
> | Scene 3 | 2| **98** |
> | Scene 4 | **12** | **12** |
> | Scene 5 | **2**| **2** |
> | Puzzle 1 | 20 | **38**|
> | Puzzle 2 | 0 | **23** |
> | Puzzle 3 | 0 | **40** |
> | Puzzle 4* | 4 | **28** |
> | Puzzle 5 | 2 | **23** |
>
> On the overlapping OGBench tasks, PCBF outperforms this uncoupled baseline on the large majority of tasks. We view this as empirical evidence that the path-coupling issue is not merely a toy-model artifact, but remains practically important in realistic large-scale MDPs as well.
>
> > The paper does not discuss the training-time computational and memory compared to baseline methods.
>
> **Response:** On OGBench (*cube-double task1* environment), PCBF uses about **59.9 GB** of GPU memory on a single **A100** with **$100\times10^4$** training steps, comparable to other flow-based baselines. Its wall-clock training time is about **$7\times10^4$ s**, compared with **$9.29\times10^4$ s** for Value Flows, **$2.07\times10^4$ s** for IFQL, **$2.18\times10^4$ s** for IQL, **$3.28\times10^4$ s** for FBRAC, and **$2.40\times10^4$ s** for FQL. PCBF is slower than non-flow baselines mainly due to repeated ODE integration.
>
> [1] Perry Dong, Chongyi Zheng, Chelsea Finn, Dorsa Sadigh, and Benjamin Eysenbach. Value Flows. International Conference on Learning Representations (ICLR), 2026.

---

> > ### Author Rebuttal · Reviewer_Qz9K · 2026-04-03
> >
> > Thank you for the clarification. I'm sorry that I will keep my score.
> >
> > Regarding discretization and coupling precision, the experiment is helpful, but it remains primarily empirical and does not fully resolve the concern about how well the coupling mechanism survives numerical errors in general settings. While the rebuttal strengthens the paper, it does not sufficiently change my main concerns, so I will keep my original score unchanged.

---

> > > ### Author Response · Authors · 2026-04-08
> > >
> > > Dear Reviewer Qz9K,
> > >
> > > We thank the reviewer for clarifying the remaining concern. We agree that empirical evidence alone may not fully resolve how well path coupling survives discretization error in general settings. We have worked hard to incorporate the reviewers’ feedback and added a **formal, environment-agnostic bound** from two perspectives:
> > >
> > > >**(i) Structural Analysis:** we demonstrated how our coupled formulation isolates and minimizes the terminal error via a structural advantage impossible for uncoupled baselines.
> > >
> > > >**(ii) Theoretical Bound (General Case):** we have also added a mathematical derivation to bound the discretization error under path coupling.
> > >
> > > **(i) Structural Analysis.**
> > > First, our claim is not that PCBF removes ODE error, but that the $\lambda$-target structurally reduces sensitivity to it. For the equation 11 in the manuscript, we could also rewrite the target as:
> > > $$
> > > u_t^\lambda = R - (1-\lambda)X_0 + (\gamma-\lambda)X' + \lambda\,v_{\theta^-}(t, Z_t^{s'} \mid s', a'),
> > > $$
> > > From the equation, we may observe that the terminal successor sample $X'$ carrying accumulated ODE integration error with coefficient $(\gamma-\lambda)$, which vanishes at $\lambda=\gamma$. By contrast, the velocity term is only a local evaluation at flow time $t$, and its perturbation is attenuated through $Z_t^{s'}=(1-t)X_0+tX'$.
> > >
> > > Crucially, this structural error attenuation is exclusive to our coupled formulation. Uncoupled baselines generate value distributions for $s$ and $s'$ using independent noises. Because their generation trajectories are disjoint, it is impossible to factor out the terminal successor state $X'$. Consequently, they are forced to absorb 100% of the accumulated ODE integration error. In contrast, our PCBF aligns these trajectories. This shared base noise is the fundamental prerequisite that allows us to isolate and scale down the $X'$ term via the **$(\gamma-\lambda)$** coefficient. This structural distinction precisely explains why PCBF exhibits robustness to ODE numerical errors compared to uncoupled methods.
> > >
> > > **(ii) General Theoretical Bound.**
> > > Let $\epsilon=\hat X'-X'$ denote the Euler discretization error where $\hat X'$ is obtained via N-step Euler integration with step size $h=1/N$, satisfying $|\epsilon|\le C_L h$ by standard ODE convergence theory. Substituting $\hat X'=X'+\epsilon$ into the $\lambda-$target, the successor interpolant shifts from $Z_t^{s'}$ to $\hat{Z}_t^{s'} = Z_t^{s'} + t\epsilon$.
> > >
> > > This give,
> > >
> > > $$
> > > \hat u_t^\lambda-u_t^\lambda=(\gamma-\lambda)\epsilon+\lambda\bigl[v_{\theta^-}(t,Z_t^{s'}+t\epsilon)-v_{\theta^-}(t,Z_t^{s'})\bigr],
> > > $$
> > >
> > > Applying the triangle inequality and the Lipschitz condition $|v_{\theta^-}(z_1,t) - v_{\theta^-}(z_2,t)| \leq L|z_1 - z_2|$, a standard regularity property established for flow-matching velocity fields under mild $\lambda$-regularity, and automatically satisfied by finite-width neural networks with bounded weights $|z_1 - z_2| = t|\epsilon|$, we obtain:
> > >
> > > $$
> > > |\hat{u}_t^\lambda - u_t^\lambda| \leq |\gamma-\lambda|\cdot|\epsilon| + \lambda \cdot L \cdot t \cdot |\epsilon| = \bigl(|\gamma-\lambda| + \lambda Lt\bigr)\cdot|\epsilon| \leq \bigl(|\gamma-\lambda| + \lambda Lt\bigr)\cdot C_L h,
> > > $$
> > >
> > > Since $t \sim \mathrm{Unif}[0,1]$ during training and $\mathbb{E}[t] = \tfrac{1}{2}$:
> > >
> > > $$
> > > \mathbb{E}_t\left[|\hat u_t^\lambda-u_t^\lambda|\right]
> > > \le
> > > \alpha(\lambda) C_L h
> > > \=
> > > \Bigl(|\gamma-\lambda|+\tfrac{\lambda L}{2}\Bigr) C_L h.
> > > $$
> > >
> > > For $\lambda \in [0,\gamma]$, the coefficient simplifies to $\alpha(\lambda) = \gamma - \lambda(1 - L/2)$. This directly yields a clear, bipartite conclusion regarding the error bound:
> > >
> > > * **When $L < 2$:** A larger $\lambda$ strictly improves the error bound, making the optimal choice $\lambda = \gamma$.
> > > * **When $L > 2$:** A larger $\lambda$ exacerbates the error, making the optimal choice $\lambda = 0$.
> > >
> > > Crucially, velocity fields learned via flow matching inherently tend to be smooth, meaning the local Lipschitz constant $L$ is typically small (often well below 2). This allows our formulation to safely and effectively leverage higher $\lambda$ values. Furthermore, even if $L$ happens to be locally large in specific, unstable regions of the state space, our framework is flexible enough to accommodate dynamic scheduling for $\lambda$, adaptively scaling it down to ensure this robustness condition is continually met. **Without shared noise**, the control variate is mathematically invalid, forcing $\lambda = 0$ and locking the error at $\alpha(0) = \gamma$ (the rigid baseline shared by all standard flow-based DRL methods). **With shared-noise coupling**, any $\lambda > 0$ (under $L < 2$) yields a strict improvement, proving that path coupling structurally and theoretically shields the target from ODE errors.
> > >
> > > **We hope that these two complementary explanations help clarify the reviewer’s concern and support a more positive evaluation of our work.**
> > >
> > > Best regards,
> > >
> > > Authors

---

### Official Review · Reviewer_Jzft · 2026-02-23

**Soundness:** 2
**Presentation:** 2
**Significance:** 2
**Originality:** 2
**Overall Recommendation:** 3
**Confidence:** 4

**Summary:**

This paper proposes Path-Coupled Bellman Flows (PCBF), a continuous-time distributional reinforcement learning framework that models return distributions via flow matching while enforcing Bellman consistency along entire flow trajectories. The key idea is to couple the generative flow paths of current and successor state-action pairs through shared base noise, inducing a geometric relationship between their velocity fields that mirrors the Bellman operator's affine structure. This coupling enables a $\lambda$-parameterized family of training targets that act as control variates, reducing variance while approximately preserving Bellman consistency. The method is evaluated on toy environments with known return distributions, OGBench, and D4RL Adroit tasks.

**Compliance With Llm Reviewing Policy:**

Affirmed.

**Final Justification:**

The problem is mainly solved, but the authors mainly promised to discuss about them in the revision. Now I consider to raise up the score to 3, and see whether it is possible to further raise up the score in the next phase.

**Key Questions For Authors:**

1. The most important problem: The author discusses too little about the related work. The section 2, Preliminaries part only discusses about Distributional RL and Flow matching, that is actually not enough. The References could also show the problem: There are only 19 papers in the reference section, and only 5 of them are after 2024. I strongly recommend the authors to polish their related work sections, especially about the previous attempts of Flow matching/diffusion models on RL, and the previous Distributional RL methods.

Here are some possible reference:
For diffusion/Flow based RL, there are:
Energy-Weighted Flow Matching (Zhang, 2025)
QGPO (Lu et al., 2023)
Diffusion-QL (Wang et al., 2022)
Diffuser (Janner et al., 2022)
SRPO (Chen et al., 2023)
IDQL (Hansen-Estruch et al., 2023)

DRL:
Ahmet Umur ¨ Ozsoy, 2025. Distributional Reinforcement Learning on Path-dependent Options
Rowland et al., 2023 — "An analysis of quantile temporal-difference learning"
Nguyen-Tang et al., 2021 — "Distributional reinforcement learning via moment matching"
Duan et al., 2021 — "Distributional soft actor-critic"
Lyle et al., 2019 — "A comparative analysis of expected and distributional reinforcement learning"
Rowland et al., 2018 — "An analysis of categorical distributional reinforcement learning
Barth-Maron et al., 2018 — D4PG

These might be some related works for DRL and Flow matching. I am not sure whether they are all directly related to the author's work, but I suggest the authors have a related work part, to discuss thoroughly about previous attempts on FM and DRL, rather than just introduce it.

2. Just from the experiment section, I do not think that the PCBF outperforms Value Flows much. For visual-cube-double-play-singletask-tasks, PCBF has really bad results. Overall, PCBF does not dominate the results. What causes these failures? Is it related to the $\lambda$ choice, the horizon length, or the nature of the return distributions in these environments? A satisfying explanation could partially address my concerns about empirical robustness.

**Limitations:**

Yes.

**Strengths And Weaknesses:**

Strength:
1. The shared-noise coupling construction (Equations 6-8) is natural and well-motivated, and the derivation of the velocity-level Bellman relation is a reasonable contribution.
2. The analysis of DCFM's incompatibility with the flow boundary condition at t=0 (Section 4.1, Proposition 4.1) is convincing. The proof that DCFM admits only constant-in-z solutions in the 1-state MRP case is clean and highlights a real structural issue.
3. The toy environments (Solitaire Dice, Bernoulli, Discrete MC Chain) with known ground-truth return distributions provide compelling evidence for distributional fidelity. The CDF comparisons in Figures 4 and 9 are particularly informative, showing clear advantages over Value Flows with increasing dcfm.
4. The authors have 38 tasks across OGBench and D4RL, with 8 seeds for state-based and 4 for visual tasks, is thorough. The ablation study on λ and the variance reduction analysis (Figure 7) are useful.

Weakness:
1. The bias analysis is incomplete and potentially misleading.
Proposition 5.7 is stated informally and only analyzed in a linear-Gaussian model with deterministic reward. This is a very restrictive setting. The paper claims shared-noise coupling makes bias "small" but the bound $O((1−\gamma)(1−t))$ only holds in this specific Gaussian case. For complex, multi-modal return distributions, which is precisely where distributional RL matters, the relationship between $X_t$ and $X'_t$ conditional $\sigma$-fields can be much more complex.
2. Figure 6 shows that performance is "highly sensitive" to $\lambda$ (the paper's own words, line 786). The optimal $\lambda$ varies dramatically across tasks (0 for visual-antmaze, 0.2-0.8 for OGBench, up to 0.99 for some D4RL tasks per Table 4). The problem is, we shall fine-tune the $\lambda$ for each different task. The paper does not provide guidance on how to set $\lambda$ adaptively (e.g., based on target network quality).
3. The experiment results seem not so good:
While PCBF shows strong distributional fidelity on toy tasks, the benchmark results are less clear-cut:
On cube-triple-play, PCBF (4±1) is dramatically worse than Value Flows (14±3).
On scene-play, PCBF (54±4) underperforms Value Flows (59±4) and CODAC (55±1).
On visual-cube-double-play, PCBF (3±0) is substantially worse than Value Flows (13±2), IQL (11±6), and FQL (6±1).
On D4RL, PCBF is competitive but does not clearly dominate.
The paper should discuss these failure cases more openly rather than claiming "strong and competitive performance across a wide variety of domains."
4. All experiments use 1D return distributions (scalar returns). The paper does not discuss or experiment with higher-dimensional return spaces, which could arise in multi-objective RL or risk-sensitive settings. The flow matching machinery is presented in general d dimensions, but all practical validation is 1D.

---

> ### Author Rebuttal · Authors · 2026-03-30
>
> We thank the reviewer for the constructive feedback. In the revision, we strengthen the theory, clarify adaptive $\\lambda$ selection, and expand the related-work section. We hope these revisions could significantly improve the clarity and positioning of the paper.
>
> > Proposition 5.7's bias analysis is too specialized.
>
> **Response:** We agree that the original bias discussion was too specialized to the linear-Gaussian case. In the revision, we replace it with a general $L^2$ bias bound that does not assume Gaussianity, unimodality, or deterministic rewards. For fixed $(s,a,t)$, let $Z = X_1' - U$ and $m_t^* = E[Z | X_t', S', A', S=s, A=a]$. Using $v_{s,a}^{(\lambda)} - v_{s,a}^* = \lambda B_{s,a}[C_{\bar v}]$ with $C_{\bar v} = \bar v(X_t', t, S', A') - Z$, and that conditional expectation is an $L^2$ contraction, we obtain a bound showing that the bias scales linearly with $|\lambda|$ and is controlled by two interpretable terms: (i) successor-model approximation error, and (ii) intrinsic geometric uncertainty in predicting the endpoint velocity from the successor interpolation. Moreover, these terms arise from an orthogonal decomposition, which yields an even sharper version of the same bound.
>
> This reformulation clarifies that our main claim is not that the general bound is always tighter than the earlier $O((1-\gamma)(1-t))$ toy estimate. Rather, the new bound is the correct assumption-light statement in the general nonlinear and multi-modal setting. We will revise the paper to make this distinction explicit and to clearly separate what is formally proven from what is heuristic.
>
> At the same time, our empirical results suggest that the shared-noise construction remains useful beyond the **Gaussian setting**. In complex nonlinear environments such as OGBench and D4RL, PCBF consistently improves over uncoupled flow-based baselines, which supports the practical value of the coupling mechanism even though the current theory does not cover the general case.
>
> > The paper does not provide guidance on how to set $\lambda$ adaptively (e.g., based on target network quality).
>
> **Response:** In PCBF, $\lambda$ acts as a control-variate coefficient, so its optimal value naturally depends on the noise level and the quality of the successor-flow prediction, which can vary across tasks and training stages.
>
> A principled way to choose $\lambda$ adaptively is therefore to estimate it from the variance-reduction objective itself. In particular, if we write the corrected target as $Y+\lambda C$, where $Y$ is the baseline sample-based target and $C$ is the control-variate term, then the variance-minimizing choice is $\lambda^\star = -\mathrm{Cov}(Y,C)/\mathrm{Var}(C)$. In practice, this could be approximated using moving-average covariance estimates or online regression. We have clarified this point in the revision about adaptive $\lambda$ selection.
>
> > The experiment results seem not so good.
>
> **Response:** We refer the reviewer to our response to reviewer Vm9o on empirical performance, where we clarify that PCBF is not claimed to uniformly dominate Value Flows, but rather to achieve competitive performance with clear strengths and weaknesses.
>
> > All experiments use 1D return distributions. The paper does not discuss or experiment with higher-dimensional return spaces, which could arise in multi-objective RL or risk-sensitive settings.
>
> **Response:**
> We agree that all experiments in the current paper are restricted to the standard scalar-return setting. Since our **primary goal** in this paper was to study whether path-coupled Bellman supervision improves flow-based distributional RL in the common benchmark  (OGBench and D4RL)  setting, we focused first on that 1D case. Conceptually, the shared-noise path-coupling construction and the $\lambda$-parameterized control-variate idea are not restricted to 1D returns, and should extend naturally to vector-valued return distributions. In the revision, we will explicitly state this as a limitation of the current paper and add discussion that extending PCBF to higher-dimensional return spaces is an important direction for future work.
>
> > The author discusses too little about the related work. The section 2, Preliminaries part only discusses about Distributional RL and Flow matching, that is actually not enough.
>
> **Response:** In revision, we expanded related work discussion in two directions. First, we added a broader discussion of generative RL, including diffusion/flow-based methods such as Diffusion-QL, QGPO, SRPO, IDQL. We clarify that our main distinction is not simply the use of flow/diffusion models, but that prior methods typically rely on endpoint-level Bellman matching  whereas PCBF uses shared-noise path coupling to align trajectories and reduce variance. Second, we expanded the discussion of the broader distributional RL literature, including D4PG, categorical/quantile DRL, moment matching, distributional SAC.

---

> > ### Author Rebuttal · Reviewer_Jzft · 2026-04-01
> >
> > Currently I think the author's rebuttal is good, so I may raise up my score to 3. However, since I still could not see the revision, I may still check about whether the author have really made a satisfying change to the paper in the phase 2 of the rebuttal, and   provide my final review.

---

> > > ### Author Response · Authors · 2026-04-08
> > >
> > > Dear Reviewer Jzft,
> > >
> > > We thank the reviewer for the response. Given the 5,000-character limit of the rebuttal, we prioritized our response to address your **two primary requests**: (i) the formal general bias-bound analysis, and (ii) the restructured version of related work.
> > >
> > > >**General $L^2$ Bias Bound.**
> > >
> > > Fix $(s,a,t)$ and let $Z:=X_1'-U$.
> > >
> > > Define
> > > $m_t^\star:=\mathbb{E}[Z\mid X_t',S',A',S=s,A=a]$.
> > >
> > > Also define $\mathcal{B}_{s,a}(f)(x,t):=\mathbb{E}[f\mid X_t=x,S=s,A=a].$
> > >
> > > Since $v_{s,a}^{(\lambda)} - v_{s,a}^\star = \lambda \cdot \mathcal{B}_{s,a}(Cv̄)$
> > >
> > > with
> > > $C_{\bar v}=\bar v(X_t',t,S',A')-Z$,
> > > we decompose
> > > $C_{\bar v}=(\bar v-m_t^\star)+(m_t^\star-Z)$.
> > >
> > > This yields the sharper current-conditioned bound
> > >
> > > \begin{equation}
> > > \lVert v_{s,a}^{(\lambda)} - v_{s,a}^{\star} \rVert_{L^2(P_{s,a})} \le |\lambda| \bigl( \varepsilon_{\mathrm{cur}}(t) + \delta_{\mathrm{cur}}(t) \bigr)
> > > \end{equation}
> > >
> > > Here $\varepsilon_{\mathrm{cur}}(t)=\lVert \mathcal{B}_{s,a}[\bar v(X_t', t, S', A') - m_t^\star](\cdot, t) \rVert$ measures successor-model error after current conditioning,
> > >
> > > while $\delta_{\mathrm{cur}}(t) := \lVert \mathcal{B}_{s,a}[m_t^\star - Z](\cdot, t) \rVert$ is the exact current-vs-successor projection mismatch.
> > >
> > > In the one-step linear-Gaussian setting, choosing $\bar v=m_t^\star$ makes the first term vanish. Under shared noise, the second term scales as $O((1-\gamma)(1-t))$.
> > >
> > > >**Added Related Work.**
> > >
> > > **Generative Models in RL.** Recent works have increasingly explored diffusion and flow models for RL planning and policy learning. For planning, Diffuser (Janner et al., 2022) introduced trajectory-level diffusion planning, where trajectories are generated by iterative denoising. For policy learning, Diffusion-QL (Wang et al., 2022) and IDQL (Hansen-Estruch et al., 2023) use diffusion models as expressive policy classes for offline RL, enabling multimodal action generation. To improve inference efficiency, SRPO (Chen et al., 2024) extracts a deterministic inference policy from pretrained diffusion behavior models, avoiding costly iterative denoising during deployment. More recently, energy-guided generation has progressed from approximate intermediate guidance to more principled formulations, including Contrastive Energy Prediction (QGPO / CEP, Lu et al., 2023) and Energy-Weighted Flow Matching (EFM, Zhang et al., 2025), which aim to make guided generation more exact and efficient. These approaches mainly focus on planning, actor learning, or guided action generation.
> > >
> > > Beyond policy and trajectory generation, generative models are increasingly being used for other RL components. For instance, EVOR (Espinosa-Dice et al., 2025) performs inference-time policy extraction using a distributional reward model learned via standard flow matching. Approach like *floq* (Agrawalla et al., 2025) have utilized flow matching to parameterize the Q-function, mapping noise to a deterministic scalar Q-value to enable iterative compute scaling. However, when extending **standard generative models** to model the full return distribution, critical bottlenecks emerge. Method like Bellman Diffusion (Li et al., 2024) attempt to align target Bellman distributions but rely on independent noise sampling. This endpoint-level Bellman matching lacks path coupling, which inevitably leads to misaligned vector fields and severe training variance.
> > >
> > > **DRL.** The DRL paradigm models the return distribution, popularized by categorical and quantile methods like C51 (Bellemare et al. 2017) and IQN (Kostrikov et al., 2021), and later scaled to continuous control via D4PG (Barth-Maron et al., 2018) and DSAC (Duan et al., 2021). Despite empirical successes, these traditional models inherently rely on discrete projections, fixed supports, or moment matching (Nguyen-Tang et al., 2021), fundamentally limiting continuous critic expressivity. While recent approaches like Value Flows (Dong et al., 2026) utilize flow matching to learn value distributions, they fail to satisfy the fundamental boundary condition at $t=0$ (i.e., exactly recovering the base prior) and struggle to maintain distributional consistency.
> > >
> > > To address the fundamental issue between Flow Matching and DRL, PCBF's formulation satisfies exact boundary conditions, ensuring rigorous Bellman consistency. To overcome the severe variance of "endpoint-level" Bellman matching, PCBF introduces Shared Base Noise. By originating current and successor flows from identical noise, we establish strict path coupling to geometrically align their trajectories. This satisfies the distributional Bellman equation smoothly across the entire path---not just endpoints---eliminating the variance explosion of uncoupled paths. Thus, PCBF complements generative actors, providing a stable, low-variance critic that enhances advanced architectures (e.g., EVOR).
> > >
> > > **We hope our responses have satisfactorily resolved your concerns. If so, we would be grateful if you could update your score accordingly.**
> > >
> > > Best regards,
> > >
> > > Authors

---

### Official Review · Reviewer_Vm9o · 2026-03-11

**Soundness:** 2
**Presentation:** 1
**Significance:** 1
**Originality:** 2
**Overall Recommendation:** 2
**Confidence:** 3

**Summary:**

This paper proposes a new variant of a continuous-time distributional RL framework, which enforces Bellman consistency along the entire flow trajectory through path coupling. They introduce the lambda-parameterized control variate to reduce the instability in the training. Experiments are conducted in some offline RL benchmarks.

**Compliance With Llm Reviewing Policy:**

Affirmed.

**Final Justification:**

The rebuttal may not change my assessment. Unifying flowing matching and distributional RL seems potential from the methodology's perspective. However, I think the benefits or motivation to do so should be highlighted and empirically demonstrated further. This paper seems to focus on the theory part, but it is still hard to follow the logic in my opinion.

**Key Questions For Authors:**

Please refer to the weakness points.

**Limitations:**

The authors have not mentioned the limitations of this study.

**Strengths And Weaknesses:**

## Strengths:
1. Incorporating flow-matching into distributional RL seems an interesting direction.
2. Theoretical results are provided to make the algorithms solid.

## Weaknesses:

1. **Unclear motivation**. This paper seems a follow-up paper to Value Flow, but it is not clear what the real challenges are they want to solve. Some general and weak motivations mentioned in the introduction are less convincing. It is very hard for me to pinpoint the contribution of this paper. In addition, I am not fully convinced by this line of research about the continuous-time distributional RL. The motivation should be strengthened in the future to help to increase the attention in the community.

2. **Writing and Theoretical Results.** It is hard to follow this paper. It is not clear what the benefits of the proposed method are over baselines. In particular, is enforcing Bellman consistency at the level of entire flow trajectories really useful in practice? More intuitions are needed to help understand this claim. The authors present Value Flows as the prior work in Section 4, but what is the purpose of introducing this section? In Section 5 about the theory, it is hard to understand the logic and why it is necessary to derive such results.

3. **Experiments.** The improvement over Value Flows is very limited in Table 1, which is not significant under such a large standard deviation. Moreover, demonstration only on the offline RL benchmark seems limited. The authors are suggested to consider online RL benchmarks.

---

> ### Author Rebuttal · Authors · 2026-03-30
>
> We thank the reviewers for the helpful feedback. In the revision, we mainly sharpen the motivation and clarify the theory and its scope.
>
> > Unclear motivation. This paper seems a follow-up paper to Value Flow, but it is not clear what the real challenges are they want to solve. The motivation should be strengthened.
>
> **Response:** We agree that our motivation requires clearer articulation. As another **reviewer WKw6** aptly highlighted in the assessment, while the field increasingly recognizes the imperative of using iterative generative models to scale value functions to complex, stochastic, and multi-modal tasks, prior attempts have fallen short. By providing a mathematically rigorous formulation, our paper is recognized as "the first to do so correctly." This observation cuts to the core of our motivation: addressing the fundamental tension in **unifying Flow Matching (FM) with Distributional RL (DRL).**
>
> Inherently, bridging these two frameworks presents a natural difficulty: FM dictates a continuous probability path that must strictly recover a fixed base prior at $t=0$, while DRL requires the target return distribution to recursively satisfy the Bellman equation.
>
> Prior attempts to link them struggle with two major structural limitations. First, they suffer from mathematical inconsistency by failing to exactly recover the prior at $t=0$, violating the fundamental requirement of FM. Second, they suffer from stochastic decoupling: by generating current and successor value distributions from independent noise, Bellman consistency is forced only at the terminal endpoints ($t=1$). This ignores the continuous geometry of the flow, leading to misaligned generation trajectories and severe training variance.
>
> PCBF is designed to systematically resolve this fundamental tension. First, it executes the FM formulation mathematically correctly by rigorously satisfying the $t=0$ boundary conditions. Beyond fixing the math, our primary conceptual breakthrough is "Path Coupling." By forcing the current and successor flows to be generated from the exact same base noise, we geometrically align their continuous trajectories. This ensures that the distributional Bellman equation is satisfied not just at the endpoints, but smoothly across the entire flow path. This path-level geometric alignment naturally derives our $\lambda$-parameterized control-variate target, transforming an inherently unstable generative process into a robust, low-variance critic for complex RL tasks.
>
> > Writing and Theoretical Results. It is hard to follow this paper. It is not clear what the benefits of the proposed method are over baselines. In particular, is enforcing Bellman consistency at the level of entire flow trajectories really useful in practice? The authors present Value Flows as the prior work in Section 4, but what is the purpose of introducing this section? In Section 5 about the theory, it is hard to understand the logic and why it is necessary to derive such results.
>
> **Response:** **Intuition and Practical Value of Pathwise Consistency:**
> Enforcing trajectory-level Bellman consistency is practically crucial. Prior flow-based methods enforce Bellman consistency **only at the terminal endpoints ($t=1$).** using independent noise, forcing the model to align unrelated trajectories and causing massive training variance. PCBF solves this via Shared Base Noise. By forcing the current and successor flows to start from the exact same random point ($t=0$), their trajectories become geometrically aligned. The model no longer struggles to match decoupled random points; instead, it stably regresses the geometric difference between these two aligned paths.
>
> **(i) Section 4 (Value Flows):** is not merely a literature review, but a formal diagnosis of the "endpoint-matching" baseline. It exposes the structural flaws of independent noise sampling (i.e., failing $t=0$ boundary conditions and high-variance decoupling), explicitly defining the problem PCBF solves. **(ii) Section 5** explains why PCBF fixes it, by characterizing the target velocity field and the bias–variance role of shared-noise coupling and the $\lambda$-target.
>
> > Experiments. The improvement over Value Flows is very limited in Table 1. Moreover, demonstration only on the offline RL benchmark seems limited.
>
> **Response:**
> PCBF is currently not fully optimized for offline RL. Value Flows includes several offline-specific stabilization tricks that we have not yet incorporated. Even so, PCBF is already competitive with Value Flows on many tasks. We will make this clearer in the revision and note that stronger offline tuning could further improve PCBF.
>
> We also agree that evaluating beyond offline RL would strengthen the paper. Our current experiments focus on offline RL because this is the most direct comparison setting with Value Flows. We will note this more explicitly in the revision, and we plan to add online or offline-to-online results in a future version.

---

> > ### Author Rebuttal · Reviewer_Vm9o · 2026-04-03
> >
> > Thank you for the response.
> >
> > Unifying flowing matching and distributional RL seems potential from the methodology's perspective. However, I think the benefits or motivation to do so should be highlighted and empirically demonstrated further. This paper seems to focus on the theory part, but it is still hard to follow the logic in my opinion. I would recommend a substantial improvement in writing and experiments, and so on, in the future submission. In summary, I keep my rating unchanged.

---

> > > ### Author Response · Authors · 2026-04-08
> > >
> > > Dear Reviewer Vm9o,
> > >
> > > Thank you again for the follow-up. We agree that the current draft does not make the geometric motivation sufficiently transparent.
> > >
> > > >**On motivation and the concrete problem addressed.**
> > >
> > > Our point is not simply that continuous-time DRL is broadly interesting. Rather, the paper studies a specific technical question that arises once return distributions are modeled with flow matching: how to impose Bellman supervision in a way that remains compatible with the fixed source boundary of flow matching and avoids the extra variance caused by independently sampled successor noise. Our contribution is therefore narrower than motivating all of continuous-time DRL: it is about how to make Bellman supervision work in a flow model in a Bellman-aligned and low-variance way.
> > >
> > > >**On how to design correct path coupling and why it is useful.**
> > >
> > > The key geometric issue can be written directly in the paper's notation. Let
> > >
> > > $$
> > > Z_t^{s^{\prime}}=(1-t) X_0+t X^{\prime}
> > > $$
> > >
> > > be the successor interpolation. A natural Bellman-looking path (used in value flow) would be
> > >
> > > $$
> > > R+\gamma Z_t^{s^{\prime}} .
> > > $$
> > >
> > > However, at $t=0$ this equals
> > >
> > > $$
> > > R+\gamma \bar{X}_0,
> > > $$
> > >
> > > not the source noise $X_0$ required by standard flow matching. This is the boundary mismatch we intended to highlight.
> > >
> > > PCBF repairs this by using
> > >
> > > $$
> > > Z_t^s=(1-t) X_0+t\left(R+\gamma X^{\prime}\right)=R+\gamma Z_t^{s^{\prime}}-\Delta_t,
> > > $$
> > >
> > > with the repair term $\Delta_t$ over the value flow to fix the boundary inconsistent.
> > >
> > > $$
> > > \Delta_t:=(1-t)\left(R-(1-\gamma) X_0\right) .
> > > $$
> > >
> > > This repaired path is exact at both boundaries:
> > >
> > > $$
> > > Z_0^s=X_0, \quad Z_1^s=R+\gamma X^{\prime} .
> > > $$
> > >
> > > So the contribution is not simply "more theory," but a source-consistent Bellman path construction: it matches the flow boundary at $t=0$ and the Bellman boundary at $t=1$, while keeping the current and successor interpolants aligned through shared base noise.
> > >
> > > This also clarifies the role of the velocity target. Differentiating the repaired path gives
> > >
> > > $$
> > > \dot{Z}_t^s=\gamma \dot{Z}_t^{s^{\prime}}+R-(1-\gamma) X_0 .
> > > $$
> > >
> > > Under linear flow matching, $\dot{Z}_t^{s^{\prime}}=X^{\prime}-X_0$, so
> > >
> > > $$
> > > \dot{Z}_t^s=R+\gamma X^{\prime}-X_0,
> > > $$
> > >
> > > which is exactly the sample Bellman target, now derived from a coupled geometry. This is why the path construction matters in practice: it yields a Bellman-aligned target without the $t=0$ inconsistency of the naive path. The $\lambda$-target then builds on top of this by using successor-velocity predictions as a control variate: $\lambda=0$ recovers the unbiased sample target, while $\lambda>0$ trades small, coupling-controlled bias for lower variance. We agree that the current draft does not explain this chain of logic clearly enough.
> > >
> > > >**On the role of Sections 4 and 5.**
> > >
> > > We also agree that the roles of Sections 4 and 5 are not communicated clearly enough. Section 4 was intended as a diagnostic comparison to Value Flows rather than a detached prior-work section to illustrate the problem mentioned above. Section 5 was intended to formalize the repaired path geometry, characterize what the $\lambda$-target actually learns, and explain the induced bias-variance tradeoff under shared-noise coupling. In revision, we will make this roadmap explicit and move less essential technical material to the appendix so that the main text foregrounds the above intuition.
> > >
> > > >**On experiments and limitations.**
> > >
> > > We agree that the current benchmark table supports a narrower claim than the paper currently makes. The strongest empirical evidence in the current submission is improved distributional fidelity and training stability; the downstream offline-RL gains are competitive but mixed, and the absence of online or offline-to-online experiments is a genuine limitation. We will state these limitations explicitly and present the toy-distribution experiments and $\lambda$-sensitivity analyses as the most direct evidence for the method.
> > >
> > > Best regards,
> > >
> > > Authors

---

### Official Review · Reviewer_WKw6 · 2026-03-13

**Soundness:** 3
**Presentation:** 3
**Significance:** 3
**Originality:** 3
**Overall Recommendation:** 5
**Confidence:** 5

**Summary:**

This paper proposes a way to use flow matching to learn the return distribution in distributional RL. They do so by leveraging the temporal difference structure and coupling the noise between the current and successor state predictions. Additionally, they propose a novel target velocity that's able to tradeoff between bootstrapping from the target network's velocity vs. the conditional flow matching path. Empirical results are demonstrated on various continuous control tasks and toy examples.

**Compliance With Llm Reviewing Policy:**

Affirmed.

**Final Justification:**

This paper provides the first correct treatment of learning return distributions via flow matching, addressing a real gap where prior work contained fundamental errors. The authors were responsive to all concerns: expanding related work, adding direct comparisons with FloQ, unifying notation, and discussing limitations. For these reasons I recommend acceptance.

**Key Questions For Authors:**

Perhaps not questions but here's some slight nit picks while I was reading the paper:
1. $\psi_\theta(X_0 \mid s, a, 1) $ and $\psi_\theta(1, X_0 \mid s, a)$ are mixed throughout the paper, e.g., section 3.1, this should be consistent.
2. It's really hard to follow the reasoning around $\lambda$ on the first read through. Although it's discussed in 3.1 it never shows up in any equations and it's hard to follow the logic towards the end of this section. Consider refactoring things so it flows more naturally and it's not as jarring every time you discuss $\lambda$ before it's actually introduced mathematically.

**Limitations:**

The authors never discuss any limitations of the method. I believe there should be discussion at least around how to deal with the magnitude of the value function, in the sense that the transport cost will increase over time as the magnitude of the value function increases over training. How should we choose the prior to deal with this? Also discussing the additional cost for evaluating the value function and how this affects performance would have been nice to see.

**Strengths And Weaknesses:**

Although there's been prior attempts at using flow matching to learn the return distribution they have all fallen short for one reason or another. The authors point out that the prior work [1] suffers from a fatal flaw in that their probability path doesn't recover the prior at t=0, one of the requirements for flow matching. I believe this work finally does this topic justice and I don't see any fundamental problems with the derivation or the method. Although this work wasn't the first to propose these ideas, it is the first to do so correctly which is the primary strength. The significance of the work is clear, iterative generative models have been getting more popular and as we continue to scale value functions to more complex, stochastic, and multi-model tasks having non-parametric models that can reliably learn the return distribution is imperative.

My main issues with the paper primarily revolve around how it's written and positioned in the literature. I believe there's many pieces of missing related work and I also have concerns about the writing, baselines, and the empirical methodology. To lay these out:
1. I believe the authors are missing a key piece of related work by not discussing [2]. This work also proposes the same coupling technique and also suggests bootstrapping from the target velocity. Granted, the primary application here is different but the framework for TD-Flow can also be used to learn the expected return through the same mixture distribution and I think it's worth discussing.
2. There is missing prior work that also attempts to learn the return distribution via flow matching [3,4] or diffusion [5] that's worth discussing and citing. Some have their own issues that may be worth pointing out if you're giving that treatment to Value Flows.
3. Many of the results exist in the literature on flow matching and make the presentation of the paper overly complex and confusing. I'd highly consider unifying the notation with [6] and citing known results to make the presentation of the paper more accessible. The novel results are the lambda-parameterization and the convergence result. Results in the paper that are direct applications include:
    - Theorem 5.5 is an application of Theorem 3 from [6] combined with two-sided conditioning in Section 4.6.3 of [6]
    - Proposition 5.6 is exactly Proposition 1 from [6]
    - Not entirely sure Section 4.1 needs to be this long, the reason value flows is broken is because the boundary conditions don't hold, you can point this out in a much more simple way using [6]
4. A typical complaint of distributional RL is that we don't leverage the return distribution for anything other than the expected value. Because of this I think it's fair game to compare with other methods that leverage flow matching to learn value functions, e.g., [7] and [2]. If you don't want to go this direction then I'd expect to see more empirical comparisons of the learnt distributions with other methods.
5. I find it shocking that C51 scores ~0 across the board in OGBench. D4PG is still a very competitive method and I assume there was no attempt at tuning the baseline methods which leaves me skeptical of the empirical results in general.

---

[1] Perry Dong, Chongyi Zheng, Chelsea Finn, Dorsa Sadigh, and Benjamin Eysenbach. Value Flows. International Conference on Learning Representations (ICLR), 2026.

[2] Jesse Farebrother, Matteo Pirotta, Andrea Tirinzoni, Rémi Munos, Alessandro Lazaric, and Ahmed Touati. Temporal Difference Flows. International Conference on Machine Learning (ICML), 2025.

[3] Nicolas Espinosa-Dice, Kiante Brantley, Wen Sun. Expressive Value Learning for Scalable Offline Reinforcement Learning. CoRR abs/2510.08218. 2025.

[4] Deshu Chen, Yuchen Liu, Zhijian Zhou, Chao Qu, Yuan Qi. Unleashing Flow Policies with Distributional Critics. CoRR abs/2509.23087. 2025.

[5] Yangming Li, Chieh-Hsin Lai, Carola-Bibiane Schönlieb, Yuki Mitsufuji, Stefano Ermon. Bellman Diffusion: Generative Modeling as Learning a Linear Operator in the Distribution Space. CoRR abs/2410.01796. 2024.

[6] Yaron Lipman, Marton Havasi, Peter Holderrieth, Neta Shaul, Matt Le, Brian Karrer, Ricky T. Q. Chen, David Lopez-Paz, Heli Ben-Hamu, Itai Gat. Flow Matching Guide and Code. CoRR abs/2412.06264. 2024.

[7] Bhavya Agrawalla, Michal Nauman, Khush Agrawal, and Aviral Kumar. floq: Training Critics via Flow-Matching for Scaling Compute in Value-Based RL. International Conference on Learning Representations (ICLR), 2026.

---

> ### Author Rebuttal · Authors · 2026-03-30
>
> We thank the reviewer for the helpful suggestions. Following your advice, we have expanded the related work and added direct empirical comparisons with prior flow-based methods.
>
> > Discussion for the missing related works.
>
> **Response:** We have comprehensively updated our Related Work to discuss these highly relevant works.
>
> **Comparison with TD-Flow [2]:** While we agree that TD-Flow and PCBF sharing technical foundations like flow matching and bootstrapping, they tackle fundamentally different problems. TD-Flow learns **Generative Horizon Models (GHMs)** (predicting future *state* distributions) using a probability mixture $P_{\text{target}} = (1-\gamma)P_{\text{real}} + \gamma P_{\text{imagined}}$. In contrast, PCBF is designed for **Value Distribution Learning** (predicting cumulative *returns*) by enforcing the Distributional Bellman Equation $Z(s,a) \overset{D}{=} r(s,a) + \gamma\, Z(s',a')$. In this algebraic transformation, $\gamma$ acts as a scaling factor.  Applying a TD-Flow's probability mixture to values would imply receiving either the immediate reward *or* future returns, violating the semantics of "cumulative" returns.
>
> **Comparison with [3,4,5]:** EVOR [3] offers a *policy-centric* design via rejection sampling, but its critic relies on standard flow matching. Similarly, DFC [4] and Bellman Diffusion [5] align target Bellman distributions using independent noise sampling. Enforcing regression between these uncoupled probability paths inevitably leads to misaligned vector fields and severe training variance. PCBF explicitly solves this. By introducing "Shared Base Noise", we geometrically align the Bellman flows and eliminate the variance explosion. Thus, PCBF is highly complementary: it provides a stable, low-variance foundational critic that can seamlessly enhance their advanced policy architectures (e.g., EVOR's actor).
>
> > Presentation and unifying notation with [6].
>
> **Response:** Following your advice, we have unified our notation with [6] and explicitly cited their foundational results to streamline our presentation and improve accessibility. We also standardized the $\psi_\theta$ flow map notation throughout , and reorganized Sec 3.1 to mathematically define $\lambda$ before its discussion, substantially improving readability.
>
> > Compare with other methods that leverage flow matching to learn value functions, e.g., [7] and [2]. If you don't want to go this direction then I'd expect to see more empirical comparisons of the learnt distributions with other methods.
>
> **Response:** We have added a direct comparison to **FloQ [7]** on the shared OGBench tasks (*cube-double-play*, *scene-play*, and *puzzle-4x4-play*, totaling 15 complex tasks) .FloQ numbers are cited directly from Table 2 of their paper. As detailed below, PCBF outperforms FloQ on **14 out of 15 tasks**.
>
> **Table: Comparison with floq on overlapping OGBench tasks.**
> | Task | floq | PCBF (Ours) |
> | :--- | :---: | :---: |
> | Cube 1 | 50±24 | **92±3** |
> | Cube 2* | 72±15 | **74±7** |
> | Cube 3 | 57±14 | **81±8** |
> | Cube 4 | 8±4 | **22±5** |
> | Cube 5 | 50±11 | **84±3** |
> | Scene 1 | **100±1** | **100±0** |
> | Scene 2* | **83±10** | 57±13 |
> | Scene 3 | **98±2** | **98±2** |
> | Scene 4 | 9±7 | **12±3** |
> | Scene 5 | 0±0 | **2±1** |
> | Puzzle 1 | **47±7** | 38±6 |
> | Puzzle 2 | 21±6 | **23±5** |
> | Puzzle 3 | 36±5 | **40±4** |
> | Puzzle 4* | 19±5 | **28±4** |
> | Puzzle 5 | 16±7 | **23±3** |
>
> **Why does PCBF outperform FloQ?** FloQ uses flow matching merely to parameterize a scalar critic, collapsing around expected $Q(s,a)$ using standard TD targets. Conversely, PCBF is a distributional method. Furthermore, PCBF's shared base noise provides pathwise Bellman-consistent targets, fundamentally reducing variance on long-horizon tasks where stochasticity compounds.
>
> **Regarding TD-Flow**: We omit TD-Flow empirically as it lacks an open-source codebase and its generative horizon objective is mathematically incompatible with scalar value learning.
>
> > I find it shocking that C51 scores ~0 across the board in OGBench. D4PG is still a very competitive method and I assume there was no attempt at tuning the baseline methods which leaves me skeptical of the empirical results in general.
>
> **Response:** The near-zero C51 scores were strictly cited from the published baselines in the Value Flows paper [1], rather than our own implementation. We have explicitly clarified this in our revised manuscript and table captions to prevent any confusion.
>
> > Limitations Discussion.
>
> **Response:** We have added a discussion in the revision.
>
> **Value Magnitude & Priors**: We agree that larger return magnitudes can make training more challenging, since flow matching must transport the base noise over a larger range.
>
> **Cost**: PCBF is more expensive than a standard scalar critic because it requires numerical integration with multiple velocity-field evaluations. We will plan to explore more advanced prior designs and more efficient implementations in future work.

---

> > ### Author Rebuttal · Reviewer_WKw6 · 2026-03-31
> >
> > I appreciate all the proposed changes and believe this paper is a valuable contribution that clears up confusion amongst the community where there are a handful of incorrect papers floating around that claim to learn the return distribution with flow matching. I recommend for acceptance.
> >
> > A couple last comments:
> >
> > - In the camera ready I'd suggest maybe excluding the C51 results, these seem obviously wrong. Perhaps replace them with the FLOQ results.
> > - I agree that TD-Flow shares a technical foundation and should be discussed in the paper. It's worth pointing out you could learn "a" return distribution with TD-Flow, it's just not the typical return distribution we think of in distributional RL. Specifically, it would be the return distribution for a transformed MDP that terminates at every timestep with probability $1 - \gamma$. Chapter 2.9 in Bellemare et al. 2023 discusses this. Perhaps discussing this nuance in detail would help clear things up for future readers.
> >
> > ---
> >
> > ### References
> >
> > Marc G. Bellemare, Will Dabney, and Mark Rowland (2023). Distributional Reinforcement Learning. MIT Press. 2023

---

> > > ### Author Response · Authors · 2026-04-01
> > >
> > > Dear Reviewer WKw6,
> > >
> > > Thank you for your support and for recommending our paper for acceptance. We are glad that our rebuttal addressed your concerns and that you find our work a valuable contribution.
> > >
> > > Regarding your suggestions for the camera-ready version, we agree and will exclude the C51 results, replacing them with the FLOQ results as suggested, and we will incorporate TD-Flow discussion to provide better clarity for future readers.
> > >
> > > We appreciate your time and constructive feedback in helping us improve the paper.
> > >
> > > Best regards,
> > >
> > > Authors

---

### Decision · Program_Chairs · 2026-04-30

**Decision:**

Accept (regular)

**Comment:**

This paper introduces a continuous-time distributional RL framework that resolves structural inconsistencies in prior flow-matching methods by coupling generative trajectories through shared base noise. The most detailed reviewer strongly championed the paper, noting it provides the first mathematically correct treatment of learning return distributions via flow matching by properly recovering the base prior at the $t=0$ boundary. While other reviewers raised initial concerns regarding restrictive linear-Gaussian bias assumptions , missing related literature, and potential numerical integration errors during discrete sampling, the authors delivered a comprehensive rebuttal. They successfully addressed these issues by introducing a general $L^2$ bias bound, expanding empirical comparisons to include baselines like FloQ, and demonstrated through both empirical residuals and theoretical bounds that their coupling mechanism inherently reduces ODE discretization errors better than uncoupled methods. Reviewers who maintained lower scores primarily cited general theory-to-practice gaps or presentation difficulties rather than identifying fundamental technical flaws. Giving appropriate weight to the high-confidence reviewer assessment and the authors' rigorous post-rebuttal improvements, the core contribution is technically solid and significant to the field. I ask the authors to take into account reviewers comments in the final version of their paper.